# Neural Stochastic Flows:
# Solver-Free Modelling and Inference for SDE Solutions

**Naoki Kiyohara**[1,2*]   **Edward Johns**[1]   **Yingzhen Li**[1]
[1]Imperial College London    [2]Canon Inc.
{n.kiyohara23, e.johns, yingzhen.li}@imperial.ac.uk

## Abstract

Stochastic differential equations (SDEs) are well suited to modelling noisy and/or irregularly-sampled time series, which are omnipresent in finance, physics, and machine learning applications. Traditional approaches require costly simulation of numerical solvers when sampling between arbitrary time points. We introduce *Neural Stochastic Flows* (NSFs) and their latent dynamic versions, which learn (latent) SDE transition laws directly using conditional normalising flows, with architectural constraints that preserve properties inherited from stochastic flow. This enables sampling between arbitrary states in a single step, providing up to two orders of magnitude speedup for distant time points. Experiments on synthetic SDE simulations and real-world tracking and video data demonstrate that NSF maintains distributional accuracy comparable to numerical approaches while dramatically reducing computation for arbitrary time-point sampling, enabling applications where numerical solvers remain prohibitively expensive.

## 1 Introduction

Stochastic differential equations (SDEs) underpin models in finance, physics, biology, and modern machine learning systems [39, 50]: they capture how a state $x_t \in \mathbb{R}^d$ evolves following a velocity field whilst being perturbed by random noise. In many real-time settings such as robots, trading algorithms, or digital twins, one often requires the *transition law* $p(x_t \mid x_s)$ describing the probability distribution of future states given earlier states over arbitrary time gaps $t - s$ [30]. Conventional approaches handle this transition law by learning neural (latent) SDEs [33] from data and simulating numerical solvers with many small steps [19, 30], which incurs high computational costs. In this regard, neural flow [4] techniques for ordinary differential equations (ODEs) bypass numerical integration by directly learning the flow map with architectural constraints and regularisation terms. However, these methods inherently cannot express stochastic dynamics required for SDE modelling. In the domain of diffusion models [12, 20, 47], a line of work leverages the associated *probability flow ODE* (PF-ODE): earlier distillation approaches compress iterative samplers by matching teacher-student trajectories under the PF-ODE [44], while consistency-style methods learn direct mappings between time points [48], with trajectory-level variants also proposed [26]. However, these techniques are tied to particular boundary conditions and diffusion processes.

An alternative approach for modelling stochastic dynamics data is through stochastic flows [32], which, under suitable conditions, describe families of strong solutions to SDEs via mappings that evolve initial states over time under shared stochasticity. These flows naturally define the transition law $p(x_t \mid x_s)$, and when parameterised by neural networks, enable direct learning of efficient one-step sampling between arbitrary time points. However, it remains an unsolved challenge for such network design, due to the requirements of satisfying several properties, e.g., identity mapping when

---

*Project page: https://nkiyohara.github.io/nsf-neurips2025/

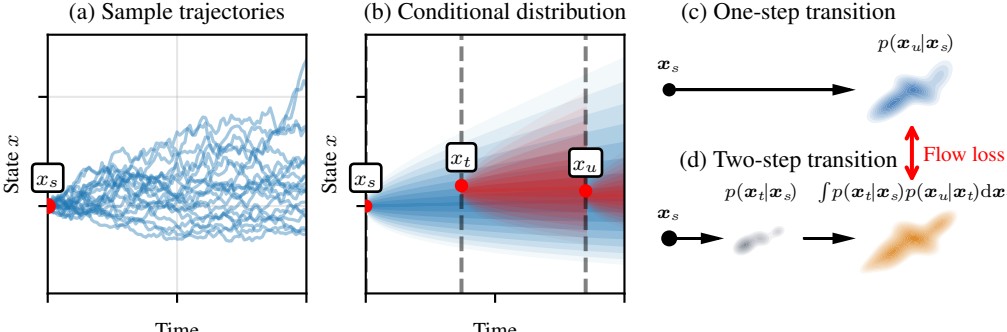

Figure 1: Comparison between (a) traditional neural SDE methods requiring numerical integration and (b) our NSF, where blue and red contours represent conditional distributions for one-step sampling and recursive application, respectively. Panels (c) and (d) illustrate our flow loss concept, ensuring distributional consistency between one-step and two-step transitions through intermediate states.

$t = s$, Markov property, and Chapman–Kolmogorov flow property that ensures self-consistency of transition laws with different number of steps.

**Contributions.** We introduce *Neural Stochastic Flows* (NSFs) to address the network design challenge of parameterising stochastic flows. NSFs employ conditional normalising flows [41, 52] to directly learn the SDE transition distributions $p(\boldsymbol{x}_t \mid \boldsymbol{x}_s)$ for any $s < t$, where the normalising flow architecture design ensures the validity of identity, Markov, and (for autonomous SDEs) stationarity properties. Specifically, given the present state $\boldsymbol{x}_s$ and elapsed time $\Delta t$, NSF draws a single Gaussian noise vector and transforms it through specially designed affine coupling layers [13, 28] that reduce to identity maps when $\Delta t := t - s = 0$ [4]. These transformations have parameters conditioned on $(\boldsymbol{x}_s, \Delta t, s)$ and scale their effect with the time interval. A bi-directional KL divergence based regularisation loss is designed to further encourage the network to satisfy Chapman–Kolmogorov flow property. Since all transformations are bijective, the transition log-density is available in closed form, allowing both training and inference to proceed without SDE solvers. A side-by-side schematic of solver-based vs. NSF sampling is shown in Fig. 1.

To summarise, Neural Stochastic Flows can:

- Learn an SDE's weak solution, in the form of a conditional distribution, directly for solver-free training and inference, with architectural constraints and loss function design to enforce desirable stochastic flow properties;

- Enable efficient one-step sampling between arbitrary time points within the trained horizon (thus suitable for modelling irregularly sampled time series), with maximum gains for distant time points;

- Be extended to noisy and/or partially observed data scenarios through latent dynamic modelling, in similar fashions as variational state-space model and latent SDEs [31, 33].

Empirically, across diverse benchmarks such as stochastic Lorenz attractor, CMU Motion Capture, and Stochastic Moving MNIST, our approach maintains distributional accuracy comparable to or better than numerical solver methods, while delivering up to two orders of magnitude faster predictions over arbitrary time intervals, with the largest gains on long-interval forecasts.

## 2 Background

**Stochastic Differential Equations.** Stochastic differential equations (SDEs) model dynamical systems subject to random perturbations:

$$\mathrm{d}\boldsymbol{x}_t = \boldsymbol{\mu}(\boldsymbol{x}_t, t)\,\mathrm{d}t + \boldsymbol{\sigma}(\boldsymbol{x}_t, t)\,\mathrm{d}\boldsymbol{W}_t, \tag{1}$$

where $\boldsymbol{x}_t \in \mathbb{R}^n$ is the state, $\boldsymbol{\mu} : \mathbb{R}^n \times \mathbb{R}_+ \to \mathbb{R}^n$ is the drift, $\boldsymbol{\sigma} : \mathbb{R}^n \times \mathbb{R}_+ \to \mathbb{R}^{n \times m}$ is the diffusion, and $\boldsymbol{W}_t$ is an $m$-dimensional Wiener process. Neural SDEs parameterise these terms with neural networks, enabling learning from data. In many applications, the conditional probability distribution

$p(\boldsymbol{x}_t \mid \boldsymbol{x}_s)$ for $s < t$ is crucial for uncertainty quantification and probabilistic forecasting. However, while numerical solvers can generate sample trajectories, they do not provide direct access to this distribution and incur computational costs that scale with the time gap $t - s$.

**Stochastic Flows of Diffeomorphisms.** An alternative perspective on SDEs comes from the theory of stochastic flows, which characterises the solution space through time-dependent mappings. For an Itô SDE as in Eq. (1) with smooth coefficients, these solutions form a *stochastic flow of diffeomorphisms* [32]. This is defined by a measurable map $\boldsymbol{\phi}_{s,t} : \mathbb{R}^n \times \Omega \to \mathbb{R}^n$ where $0 < s < t < +\infty$ and $\Omega$ is the sample space, satisfying:

1. **Diffeomorphism**: For any $s < t$ and $\boldsymbol{\omega} \in \Omega$, the map $\boldsymbol{x} \mapsto \boldsymbol{\phi}_{s,t}(\boldsymbol{x}, \boldsymbol{\omega})$ is almost surely a diffeomorphism.

2. **Independence**: Maps $\boldsymbol{\phi}_{t_1,t_2}, \ldots, \boldsymbol{\phi}_{t_{n-1},t_n}$ for any non-overlapping sequence $0 \leq t_1 \leq t_2 \leq \ldots \leq t_n$ are independent, which yields the Markov property of the flow.

3. **Flow property**: For any $0 \leq t_1 \leq t_2 \leq t_3$, any point $\boldsymbol{x} \in \mathbb{R}^n$, and any $\boldsymbol{\omega} \in \Omega$:
$$\boldsymbol{\phi}_{t_1,t_3}(\boldsymbol{x}, \boldsymbol{\omega}) = \boldsymbol{\phi}_{t_2,t_3}\big(\boldsymbol{\phi}_{t_1,t_2}(\boldsymbol{x}, \boldsymbol{\omega}), \boldsymbol{\omega}\big).$$

4. **Identity property**: For any $t \geq 0$, any point $\boldsymbol{x} \in \mathbb{R}^n$, and any $\boldsymbol{\omega} \in \Omega$: $\boldsymbol{\phi}_{t,t}(\boldsymbol{x}, \boldsymbol{\omega}) = \boldsymbol{x}$.

Furthermore, when the SDE is autonomous, where the drift and diffusion terms are independent of time, an additional property holds:

5. **Stationarity**: For any $s \leq t$, $r \geq 0$, and $\boldsymbol{\omega} \in \Omega$, the maps $\boldsymbol{\phi}_{s,t}(\boldsymbol{x}, \boldsymbol{\omega})$ and $\boldsymbol{\phi}_{s+r,t+r}(\boldsymbol{x}, \boldsymbol{\omega})$ have the same probability distribution.

Solver-based neural SDEs (also see Section 5) require costly numerical integration. Leveraging the flow properties formalised above, in the next section we introduce a solver-free alternative, NSFs, as efficient models for stochastic systems that provide direct access to transition probability distributions of SDE-governed processes.

## 3 Neural Stochastic Flows

While stochastic flows of diffeomorphisms provide a powerful theoretical framework for understanding strong solutions of SDEs, they require modelling infinite-dimensional sample paths, which is computationally intractable. Instead, in this section we focus on developing neural network architectures for modelling the probability distributions that characterise weak solutions of SDEs, by deriving appropriate conditions and loss functions for the network using properties of stochastic flows.

For an SDE, the relationship between a strong solution (represented by a stochastic flow $\phi$) to its corresponding weak solution (represented by a conditional probability distribution) is expressed as:

$$p(\boldsymbol{x}_{t_j} \mid \boldsymbol{x}_{t_i}; t_i, t_j - t_i) := \int_{\Omega} \delta\left(\boldsymbol{x}_{t_j} - \boldsymbol{\phi}_{t_i,t_j}(\boldsymbol{x}_{t_i}, \boldsymbol{\omega})\right) p(\boldsymbol{\omega}) \, \mathrm{d}\boldsymbol{\omega}, \tag{2}$$

where $\delta(\cdot)$ denotes the Dirac delta function and $\Omega$ represents the infinite-dimensional sample space containing complete Brownian motion trajectories. Instead of directly modelling the map $\phi$, we approximate the resulting probability distribution $p_{\boldsymbol{\theta}}(\boldsymbol{x}_{t_j} \mid \boldsymbol{x}_{t_i}; t_i, t_j - t_i) \approx p(\boldsymbol{x}_{t_j} \mid \boldsymbol{x}_{t_i}; t_i, t_j - t_i)$ using a *Neural Stochastic Flow* (NSF), a parametric model $\boldsymbol{f}_{\boldsymbol{\theta}}$ that transforms Gaussian samples $\boldsymbol{\varepsilon}$ into the desired distributions via conditional normalising flows:

$$p_{\boldsymbol{\theta}}(\boldsymbol{x}_{t_j} \mid \boldsymbol{x}_{t_i}; t_i, t_j - t_i) = \int \delta\big(\boldsymbol{x}_{t_j} - \boldsymbol{f}_{\boldsymbol{\theta}}(\boldsymbol{x}_{t_i}, t_i, t_j - t_i, \boldsymbol{\varepsilon})\big) \mathcal{N}(\boldsymbol{\varepsilon} \mid \boldsymbol{0}, \boldsymbol{I}) \, \mathrm{d}\boldsymbol{\varepsilon}, \tag{3}$$

Therefore $\boldsymbol{f}_{\boldsymbol{\theta}}$ serves as our parametric model to approximate the SDE's weak solution.

**Conditions.** Under the NSF framework, in below we reformulate the conditions of stochastic flows as strong solutions of SDEs to conditions of NSFs as weak solutions:

1. **Independence**: For any sequence $0 \leq t_1 \leq t_2 \leq \ldots \leq t_n$, the conditional probabilities $p_{\boldsymbol{\theta}}(\boldsymbol{x}_{t_2} \mid \boldsymbol{x}_{t_1}; t_1, t_2 - t_1), \ldots, p_{\boldsymbol{\theta}}(\boldsymbol{x}_{t_n} \mid \boldsymbol{x}_{t_{n-1}}; t_{n-1}, t_n - t_{n-1})$ must be independent.

2. **Flow property**: For any $0 \leq t_i \leq t_j \leq t_k$, the joint distribution must satisfy the Chapman–Kolmogorov equation:

$$p_{\boldsymbol{\theta}}(\boldsymbol{x}_{t_k} \mid \boldsymbol{x}_{t_i}; t_i, t_k - t_i) = \int p_{\boldsymbol{\theta}}(\boldsymbol{x}_{t_k} \mid \boldsymbol{x}_{t_j}; t_j, t_k - t_j) p_{\boldsymbol{\theta}}(\boldsymbol{x}_{t_j} \mid \boldsymbol{x}_{t_i}; t_i, t_j - t_i) \, \mathrm{d}\boldsymbol{x}_{t_j}. \tag{4}$$

3. **Identity property**: At the same time $t$, the distribution must reduce to a delta function, indicating no change: $p_{\boldsymbol{\theta}}(\boldsymbol{x}_{t_i} = \boldsymbol{x} \mid \boldsymbol{x}_{t_i}; t_i, 0) = \delta(\boldsymbol{x} - \boldsymbol{x}_{t_i})$.

For autonomous SDEs, we restate the stationarity condition in terms of our parametric model:

4. **Stationarity**: The conditional distributions must exhibit stationarity such that for any $t_i, t_j, r \geq 0$, $p_{\boldsymbol{\theta}}(\boldsymbol{x}_{t_j} \mid \boldsymbol{x}; t_i, t_j - t_i) = p_{\boldsymbol{\theta}}(\boldsymbol{x}_{t_j+r} \mid \boldsymbol{x}; t_i + r, t_j - t_i)$.

### 3.1 Conditional Normalising Flow Design

Let $\boldsymbol{c} := (\boldsymbol{x}_{t_i}, \Delta t, t_i)$ denote the conditioning parameters, where $t_i$ is included only for non-autonomous systems and omitted otherwise, and let $\Delta t := t_j - t_i$. We instantiate the sampling procedure of the flow distribution (Eq. (3)) as

$$z = \underbrace{\boldsymbol{x}_{t_i} + \Delta t \cdot \mathrm{MLP}_\mu(\boldsymbol{c}; \boldsymbol{\theta}_\mu)}_{\boldsymbol{\mu}(\boldsymbol{c})} + \underbrace{\sqrt{\Delta t} \cdot \mathrm{MLP}_\sigma(\boldsymbol{c}; \boldsymbol{\theta}_\sigma)}_{\boldsymbol{\sigma}(\boldsymbol{c})} \odot \boldsymbol{\varepsilon}, \quad \boldsymbol{\varepsilon} \sim \mathcal{N}(\boldsymbol{0}, \boldsymbol{I}), \tag{5}$$

$$\boldsymbol{x}_{t_j} = \boldsymbol{f}_{\boldsymbol{\theta}}(z, \boldsymbol{c}) = \boldsymbol{f}_L(\cdot; \boldsymbol{c}, \boldsymbol{\theta}_L) \circ \boldsymbol{f}_{L-1}(\cdot; \boldsymbol{c}, \boldsymbol{\theta}_{L-1}) \circ \cdots \circ \boldsymbol{f}_1(z; \boldsymbol{c}, \boldsymbol{\theta}_1). \tag{6}$$

Our architecture integrates a parametric Gaussian initialisation with a sequence of bijective transformations. The state-dependent Gaussian, centred at $\boldsymbol{\mu}(\boldsymbol{c})$ with noise scale $\boldsymbol{\sigma}(\boldsymbol{c})$, follows the similar form to the Euler–Maruyama discretisation with drift scaled by $\Delta t$ and diffusion by $\sqrt{\Delta t}$. The subsequent bijective transformations $\boldsymbol{f}_1$ through $\boldsymbol{f}_L$ are implemented as conditioned coupling flows [4, 13, 28] whose parameters depend on $\boldsymbol{c}$. Each layer splits the state $z$ into two partitions $(\boldsymbol{z}_\mathrm{A}, \boldsymbol{z}_\mathrm{B})$ and applies an affine update to one partition conditioned on the other and on $\boldsymbol{c}$:

$$\boldsymbol{f}_i(z; \boldsymbol{c}, \boldsymbol{\theta}_i) = \mathrm{Concat}\Big(\boldsymbol{z}_\mathrm{A}, \boldsymbol{z}_\mathrm{B} \odot \exp\big(\Delta t \, \mathrm{MLP}_\mathrm{scale}^{(i)}(\boldsymbol{z}_\mathrm{A}, \boldsymbol{c}; \boldsymbol{\theta}_\mathrm{scale}^{(i)})\big) + \Delta t \, \mathrm{MLP}_\mathrm{shift}^{(i)}(\boldsymbol{z}_\mathrm{A}, \boldsymbol{c}; \boldsymbol{\theta}_\mathrm{shift}^{(i)})\Big), \tag{7}$$

with alternating partitions across layers. The explicit $\Delta t$ factor ensures $\boldsymbol{f}_i(z; \boldsymbol{c}, \boldsymbol{\theta}_i) = z$ when $\Delta t = 0$ and, combined with the form of the base Gaussian (Eq. (5)), preserves the identity at zero time gap. Stacking such layers yields an expressive diffeomorphism [49] while keeping the Jacobian log-determinant tractable for loss computation. Independence property is ensured by the conditional sampling on the initial state $\boldsymbol{x}_{t_i}$ without any overlap between the transitions, and stationarity is obtained by omitting $t_i$ from $\boldsymbol{c}$ when modelling autonomous SDEs.

### 3.2 A Regularisation Loss for Flow Property

To train an NSF, apart from using supervised learning loss functions based on e.g., maximum likelihood, we add an additional discrepancy term $\mathcal{L}_\mathrm{flow}$ that encourages the flow property as formalised in Eq. (4). We require a measure of mismatch between the one-step distribution $p_{\boldsymbol{\theta}}(\boldsymbol{x}_{t_k} \mid \boldsymbol{x}_{t_i})$ and the two-step marginal $\int p_{\boldsymbol{\theta}}(\boldsymbol{x}_{t_k} \mid \boldsymbol{x}_{t_j}) \, p_{\boldsymbol{\theta}}(\boldsymbol{x}_{t_j} \mid \boldsymbol{x}_{t_i}) \, \mathrm{d}\boldsymbol{x}_{t_j}$. Several candidates exist such as optimal-transport/Wasserstein, Stein, adversarial, and kernel MMD. Among them, we choose KL divergences because we can minimise the upper-bounds of the KL divergences in a tractable way, and since KL is asymmetric, we use both directions: the forward KL promotes coverage, while the reverse KL penalises unsupported regions. Direct KLs remain intractable due to the marginalisation over intermediate state $\boldsymbol{x}_{t_j}$ in two-step side. Therefore, we introduce a bridge distribution $b_{\boldsymbol{\xi}}(\boldsymbol{x}_{t_j} \mid \boldsymbol{x}_{t_i}, \boldsymbol{x}_{t_k})$ as an auxiliary variational distribution [1, 40, 45] and optimise variational upper bounds for both directions (Appendix B).

Specifically, for the forward KL divergence:

$$D_{\mathrm{KL}}\left(p_{\boldsymbol{\theta}}\left(\boldsymbol{x}_{t_k} \mid \boldsymbol{x}_{t_i}\right) \,\middle\|\, \int p_{\boldsymbol{\theta}}\left(\boldsymbol{x}_{t_k} \mid \boldsymbol{x}_{t_j}\right) p_{\boldsymbol{\theta}}\left(\boldsymbol{x}_{t_j} \mid \boldsymbol{x}_{t_i}\right) \mathrm{d}\boldsymbol{x}_{t_j}\right)$$

$$\leq \mathop{\mathbb{E}}_{\boldsymbol{x}_{t_k} \sim p_{\boldsymbol{\theta}}(\cdot \mid \boldsymbol{x}_{t_i})}\left[\mathop{\mathbb{E}}_{\boldsymbol{x}_{t_j} \sim b_{\boldsymbol{\xi}}(\cdot \mid \boldsymbol{x}_{t_i}, \boldsymbol{x}_{t_k})}\left[\log \frac{p_{\boldsymbol{\theta}}\left(\boldsymbol{x}_{t_k} \mid \boldsymbol{x}_{t_i}\right) b_{\boldsymbol{\xi}}\left(\boldsymbol{x}_{t_j} \mid \boldsymbol{x}_{t_i}, \boldsymbol{x}_{t_k}\right)}{p_{\boldsymbol{\theta}}\left(\boldsymbol{x}_{t_k} \mid \boldsymbol{x}_{t_j}\right) p_{\boldsymbol{\theta}}\left(\boldsymbol{x}_{t_j} \mid \boldsymbol{x}_{t_i}\right)}\right]\right]$$

$$=: \mathcal{L}_{\mathrm{flow,\ 1\text{-}to\text{-}2}}\left(\boldsymbol{\theta}, \boldsymbol{\xi}; t_i, t_j, t_k\right). \tag{8}$$

And for the reverse KL divergence:

$$D_{\mathrm{KL}}\left(\int p_{\boldsymbol{\theta}}\left(\boldsymbol{x}_{t_k} \mid \boldsymbol{x}_{t_j}\right) p_{\boldsymbol{\theta}}\left(\boldsymbol{x}_{t_j} \mid \boldsymbol{x}_{t_i}\right) \mathrm{d}\boldsymbol{x}_{t_j} \,\middle\|\, p_{\boldsymbol{\theta}}\left(\boldsymbol{x}_{t_k} \mid \boldsymbol{x}_{t_i}\right)\right)$$

**Algorithm 1:** Optimisation procedure for (Latent) Neural Stochastic Flows
Follow ♦ for NSF, ♠ for Latent NSF; lines without markers are run for both models.

**Input:** Transition dataset $\mathcal{D}$; learning rates $\eta_\theta, \eta_\xi$ (and $\eta_\phi, \eta_\psi$ for latent); inner steps $K$
**Output:** Trained parameters $\boldsymbol{\theta}, \boldsymbol{\xi}$ (and $\phi, \psi$ for latent)

**while** *not converged* **do**
    Sample a minibatch from $\mathcal{D}$
    ♠ Encode sampled data into latent states using Eq. (15)
    Sample time triplets $(t_i, t_j, t_k)$ with $t_i < t_j < t_k$ (see Appendix B for details)
    **for** $k = 1$ **to** $K$ **do**
        Compute $\mathcal{L}_{\text{flow}}(\boldsymbol{\theta}, \boldsymbol{\xi})$ using Eq. (10)
        Update bridge parameters: $\boldsymbol{\xi} \leftarrow \boldsymbol{\xi} - \eta_\xi \nabla_{\boldsymbol{\xi}} \mathcal{L}_{\text{flow}}$
    ♦ Compute total loss $\mathcal{L}(\boldsymbol{\theta}, \boldsymbol{\xi})$ using Eq. (11)
    ♠ Compute total loss $\mathcal{L}(\boldsymbol{\theta}, \boldsymbol{\xi}, \phi, \psi)$ using Eq. (18)
    ♦ Update NSF parameters: $\boldsymbol{\theta} \leftarrow \boldsymbol{\theta} - \eta_\theta \nabla_{\boldsymbol{\theta}} \mathcal{L}$
    ♠ Update NSF/encoder/decoder parameters: $\{\boldsymbol{\theta}, \phi, \psi\} \leftarrow \{\boldsymbol{\theta}, \phi, \psi\} - \eta_{\{\theta, \phi, \psi\}} \nabla_{\{\theta, \phi, \psi\}} \mathcal{L}$

$$
\leq \mathop{\mathbb{E}}_{\boldsymbol{x}_{t_j} \sim p_{\boldsymbol{\theta}}(\cdot | \boldsymbol{x}_{t_i})} \left[ \mathop{\mathbb{E}}_{\boldsymbol{x}_{t_k} \sim p_{\boldsymbol{\theta}}(\cdot | \boldsymbol{x}_{t_j})} \left[ \log \frac{p_{\boldsymbol{\theta}}\left(\boldsymbol{x}_{t_j} \mid \boldsymbol{x}_{t_i}\right) p_{\boldsymbol{\theta}}\left(\boldsymbol{x}_{t_k} \mid \boldsymbol{x}_{t_j}\right)}{b_{\boldsymbol{\xi}}\left(\boldsymbol{x}_{t_j} \mid \boldsymbol{x}_{t_i}, \boldsymbol{x}_{t_k}\right) p_{\boldsymbol{\theta}}\left(\boldsymbol{x}_{t_k} \mid \boldsymbol{x}_{t_i}\right)} \right] \right]
$$
$$
=: \mathcal{L}_{\text{flow, 2-to-1}}\left(\boldsymbol{\theta}, \boldsymbol{\xi}; t_i, t_j, t_k\right). \tag{9}
$$

The flow loss $\mathcal{L}_{\text{flow}}$ combines the above two terms, balancing the consistency in both directions:

$$
\mathcal{L}_{\text{flow}}(\boldsymbol{\theta}, \boldsymbol{\xi}) = \mathop{\mathbb{E}}_{p(t_i, t_j, t_k)} \left[ \mathcal{L}_{\text{flow, 1-to-2}}(\boldsymbol{\theta}, \boldsymbol{\xi}; t_i, t_j, t_k) + \mathcal{L}_{\text{flow, 2-to-1}}(\boldsymbol{\theta}, \boldsymbol{\xi}; t_i, t_j, t_k) \right], \tag{10}
$$

The total loss for training NSFs combines the negative log-likelihood objective with the flow loss:

$$
\mathcal{L}(\boldsymbol{\theta}, \boldsymbol{\xi}) = - \mathop{\mathbb{E}}_{(\boldsymbol{x}_{t_i}, \boldsymbol{x}_{t_j}, t_i, t_j) \sim \mathcal{D}} \left[ \log p_{\boldsymbol{\theta}}(\boldsymbol{x}_{t_j} \mid \boldsymbol{x}_{t_i}; t_i, t_j - t_i) \right] + \lambda \mathcal{L}_{\text{flow}}(\boldsymbol{\theta}, \boldsymbol{\xi}), \tag{11}
$$

where $\mathcal{D}$ represents the dataset and $\lambda$ is a hyperparameter that controls the strength of the flow consistency constraint. The model parameters $\boldsymbol{\theta}$ and bridge model parameters $\boldsymbol{\xi}$ are optimised using gradient-based methods; a procedure is summarised in Algorithm 1 (see Appendix C.1 for details).

## 4 Latent Neural Stochastic Flows

Irregularly-sampled real-world sequences seldom expose the full system state; instead we observe noisy measurements $\boldsymbol{o}_{t_i}$ at times $0 = t_0 < t_1 < \cdots < t_T$. To model the hidden continuous-time dynamics while remaining solver-free, we introduce the *Latent Neural Stochastic Flows* (Latent NSFs) as variational state-space models (VSSMs) whose transition kernels are governed by NSFs.

### 4.1 Recap: Variational State-Space Models (VSSMs)

A VSSM [8, 21, 22, 31] extends the variational autoencoder framework [29, 42] to sequential data by endowing a sequence of latent states $\boldsymbol{x}_{0:T}$ with Markovian dynamics while explaining the observations $\boldsymbol{o}_{0:T}$ through a conditional emission process. The joint distribution factorises as

$$
p_{\boldsymbol{\theta}, \boldsymbol{\psi}}(\boldsymbol{x}_{0:T}, \boldsymbol{o}_{0:T}) = p_{\boldsymbol{\theta}}(\boldsymbol{x}_0) \prod_{t=1}^{T} p_{\boldsymbol{\theta}}(\boldsymbol{x}_t \mid \boldsymbol{x}_{t-1}) p_{\boldsymbol{\psi}}(\boldsymbol{o}_t \mid \boldsymbol{x}_t), \tag{12}
$$

where each conditional is typically Gaussian with its distributional parameters produced by neural networks. Amortised inference is carried out using a Gaussian encoder $q_\phi(\boldsymbol{x}_t \mid \boldsymbol{x}_{t-1}, \boldsymbol{o}_t)$ for every time-step $t$, and the encoder and decoder are jointly trained using the negative evidence lower bound (NELBO) loss, also known as the variational free energy (VFE):

$$
\mathcal{L}(\boldsymbol{\theta}, \boldsymbol{\psi}, \phi) = \mathop{\mathbb{E}}_{q_\phi} \left[ -\log p_{\boldsymbol{\theta}}(\boldsymbol{x}_0) - \sum_{t=1}^{T} \left[ \log p_{\boldsymbol{\theta}}(\boldsymbol{x}_t \mid \boldsymbol{x}_{t-1}) + \log p_{\boldsymbol{\psi}}(\boldsymbol{o}_t \mid \boldsymbol{x}_t) - \log q_\phi(\boldsymbol{x}_t \mid \boldsymbol{x}_{t-1}, \boldsymbol{o}_t) \right] \right]. \tag{13}
$$

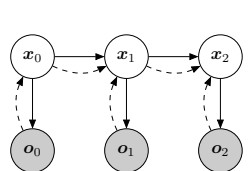 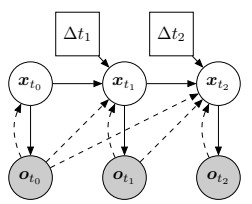 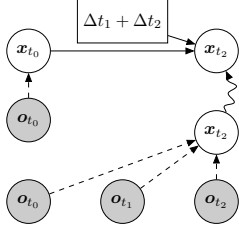

(a) Variational State-Space Model      (b) Latent Neural Stochastic Flow      (c) Skip KL Divergence

Figure 2: Graphical models. (a) Standard state-space model with discrete-time transitions between states. (b) NSF model with continuous-time transitions parameterised by time intervals ($\Delta t_i :=$ $t_i - t_{i-1}$). (c) Example of skip KL divergence over two time steps. The upper row shows the two-step-ahead prediction from the posterior at $t_0$ via NSF, while the lower row shows the posterior at $t_2$ conditioned on all observations $(\boldsymbol{o}_{t_0}, \boldsymbol{o}_{t_1}, \boldsymbol{o}_{t_2})$. Across all models: solid arrows represent generative processes, dashed arrows represent inference processes, wavy arrows in (c) represent KL divergences.

### 4.2 Neural Stochastic Flows as Latent Transition Kernels

To model irregularly-sampled time series, we require the latent dynamics to evolve in continuous-time. We therefore implement the transition distribution $p_{\boldsymbol{\theta}}(\boldsymbol{x}_t \mid \boldsymbol{x}_{t-1})$ in (12) with an NSF $p_{\boldsymbol{\theta}}(\boldsymbol{x}_{t_j} \mid \boldsymbol{x}_{t_i}; t_{i-1}, \Delta t_i)$ which parameterises the weak solution of an SDE as follows. As illustrated in Fig. 2(a), a standard VSSM uses discrete-time transitions, whereas latent NSFs in Fig. 2(b) incorporates continuous-time dynamics through time intervals.

**Generative model.** Let $\boldsymbol{x}_{t_i} \in \mathbb{R}^d$ be the latent state and $\Delta t_i := t_i - t_{i-1}$. The joint distribution factorises as

$$p_{\boldsymbol{\theta},\boldsymbol{\psi}}(\boldsymbol{x}_{t_{0:T}}, \boldsymbol{o}_{t_{0:T}}) = p_{\boldsymbol{\theta}}(\boldsymbol{x}_{t_0}) \prod_{i=1}^{T} p_{\boldsymbol{\theta}}(\boldsymbol{x}_{t_i} \mid \boldsymbol{x}_{t_{i-1}}; t_{i-1}, \Delta t_i) \, p_{\boldsymbol{\psi}}(\boldsymbol{o}_{t_i} \mid \boldsymbol{x}_{t_i}), \qquad (14)$$

where $p_{\boldsymbol{\theta}}(\boldsymbol{x}_{t_i} \mid \boldsymbol{x}_{t_{i-1}}; \cdot)$ is an NSF, hence supports arbitrary $\Delta t_i$ without numerical integration.

**Variational posterior.** For the inference model, we employ a recurrent neural network encoder that processes the observation sequence [7]:

$$q_{\boldsymbol{\phi}}(\boldsymbol{x}_{t_{0:T}} \mid \boldsymbol{o}_{\leq t_T}) = \prod_{i=0}^{T} \mathcal{N}(\boldsymbol{x}_{t_i} \mid \boldsymbol{m}_{t_i}, \mathrm{diag}(\boldsymbol{s}_{t_i}^2)), \quad (\boldsymbol{m}_{t_i}, \boldsymbol{s}_{t_i}) = \mathrm{GRU}_{\boldsymbol{\phi}}([\boldsymbol{o}_{t_i}, \Delta t_i, t_i], \boldsymbol{h}_{t_{i-1}}),$$
$$(15)$$

with $\boldsymbol{h}_{t_{-1}} = \boldsymbol{0}$, and the absolute time ($t_{i-1}$ in Eq. (14) and $t_i$ in Eq. (15)) is only included for non-autonomous systems and omitted when modelling autonomous SDEs.

**Learning objective.** Our goal is to train generative model $p_{\boldsymbol{\theta},\boldsymbol{\psi}}(\boldsymbol{x}_{t_{0:T}}, \boldsymbol{o}_{t_{0:T}})$ and inference model $q_{\boldsymbol{\phi}}(\boldsymbol{x}_{t_{0:T}} \mid \boldsymbol{o}_{\leq t_T})$. A standard choice is the $\beta$-weighted negative ELBO ($\beta$-NELBO) loss [18]:

$$\mathcal{L}_{\beta\text{-NELBO}} = \sum_{i=0}^{T} \Big[ - \underset{q_{\boldsymbol{\phi}}}{\mathbb{E}} \big[ \log p_{\boldsymbol{\psi}}(\boldsymbol{o}_{t_i} \mid \boldsymbol{x}_{t_i}) \big] + \beta D_{\mathrm{KL}}\big( q_{\boldsymbol{\phi}}(\boldsymbol{x}_{t_i} \mid \boldsymbol{o}_{\leq t_i}) \,\big\|\, p_{\boldsymbol{\theta}}(\boldsymbol{x}_{t_i} \mid \boldsymbol{x}_{t_{i-1}}) \big) \Big], \qquad (16)$$

which focuses only on adjacent time steps, potentially leading to the error accumulation for long-term dependencies. To address this, we propose to add a skip-ahead KL divergence loss (Fig. 2(c)):

$$\mathcal{L}_{\mathrm{skip}} = \sum_{i=0}^{T-\tau} \underset{j \sim \mathcal{U}\{i+2, \tau\}}{\mathbb{E}} \big[ D_{\mathrm{KL}}\big( q_{\boldsymbol{\phi}}(\boldsymbol{x}_{t_j} \mid \boldsymbol{o}_{\leq t_j}) \,\big\|\, p_{\boldsymbol{\theta}}(\boldsymbol{x}_{t_j} \mid \boldsymbol{x}_{t_i}) \big) \big]. \qquad (17)$$

Unlike traditional overshooting methods [16] that require recursive transitions, latent NSF enables direct sampling across arbitrary time gaps, enabling this objective to be computed efficiently.

Combining the above two losses and the flow loss (Eq. (10)), we get the total loss:

$$\mathcal{L}_{\mathrm{total}} = \mathcal{L}_{\beta\text{-NELBO}} + \lambda \, \mathcal{L}_{\mathrm{flow}} + \beta_{\mathrm{skip}} \mathcal{L}_{\mathrm{skip}}, \qquad (18)$$

where $\lambda$ and $\beta_{\mathrm{skip}}$ are hyperparameters that control the strength of the flow loss and skip-ahead KL divergence loss, respectively.

A training procedure for the latent model is summarised in Algorithm 1 (see Appendix C.2).

# 5 Related Work

We categorise prior work in terms of whether learning and/or sampling avoid fine-grained numerical time-stepping, and the class of dynamics targeted, as summarised in Table 1.

**Modelling general ODEs.** Neural ODEs [6] learn a parametric vector field that is integrated by a numerical solver at both training and test time. Their runtime therefore scales with the number of function evaluations required by the solver. Neural flows [4] side-step this cost by learning the solution map directly.

**Modelling prescribed SDEs/PF-ODEs.** Score-based generative models [47] and flow matching [34] are continuous-time generative models that learn reverse SDEs/PF-ODEs of boundary-conditioned diffusion processes. Fast generation methods have been developed through learning direct mappings [26, 48] or vector-field straightening [34, 53]. However, they are specifically designed for prescribed boundary-conditioned diffusion processes, and do not provide a general transition density for arbitrary SDEs.

**Modelling general SDEs.** Neural (latent) SDEs [25, 33, 38, 50] utilise neural networks to model the drift and diffusion terms of an SDE and approximate its trajectories

Table 1: Solver requirement across continuous-time differential equation models. 'Pre-defined diffusion SDEs/ODEs' refers to a boundary-conditioned diffusion process bridging a fixed base distribution and the data distribution utilised in diffusion models.

| Method(s) | Target dynamics | Solver-free training | Solver-free inference |
|---|---|---|---|
| Neural ODEs [6] | General ODEs | ✗ | ✗ |
| Neural flows [4] | General ODEs | ✓ | ✓ |
| Score-based diffusion via reverse SDEs/PF-ODEs [47]; flow matching [34] | Pre-defined diffusion SDEs/ODEs | ✓ | ✗ |
| Progressive distillation [44] | Pre-defined diffusion SDEs/ODEs | ✗ | ✓ |
| Consistency models [26, 48]; rectified flows [35] | Pre-defined diffusion SDEs/ODEs | ✓ | ✓ |
| Neural (latent) SDEs [25, 33, 38, 46, 50] | General Itô SDEs | ✗ | ✗ |
| ARCTA [9]; SDE matching [3] | General Itô SDEs | ✓ | ✗ |
| **Neural Stochastic Flows** | General Itô SDEs | ✓ | ✓ |

via stochastic solvers, whose computational cost grows linearly with the prediction horizon. Recent advances have aimed to mitigate this cost by improving sampling efficiency [46] or by introducing solver-free approaches such as ARCTA [9] and SDE matching [3], which learn latent SDEs by modelling posterior marginals and aligning them to the prior dynamics. However, all these methods remain solver-dependent at inference time.

**Position of the present work.** NSF unifies the solver-free philosophy of neural flows with the expressive power of neural (latent) SDEs. It learns the Markov transition density as a conditional normalising flow, which satisfies the conditions of weak solutions of SDEs by design and regularisation objectives. Unlike diffusion model-specific accelerations, NSF handles arbitrary Itô SDEs and extends naturally to latent sequence models. The resulting method removes numerical integration while retaining closed-form likelihoods.

# 6 Experiments

We evaluate NSF and latent NSF on three diverse tasks: modelling a synthetic stochastic Lorenz attractor [33, 36], predicting real-world human motion (CMU Motion Capture [15]), and generative video modelling (Stochastic Moving MNIST [11]). Our goal is to show that (latent) NSF matches or improves upon the accuracy (task-specific metrics) of solver-based neural SDEs while drastically reducing computational cost in terms of FLOPs and runtime. Full details are in Appendix E.

## 6.1 Stochastic Lorenz Attractor

We first test NSF on the stochastic Lorenz system [33, 36], a standard chaotic dynamics benchmark. We use the setup from Li et al. [33], generating 1,024 training trajectories and comparing against baselines including latent SDE [33], SDE matching [3], and Stable Neural SDE variants (Neural LSDE, Neural GSDE, Neural LNSDE) [38] as baselines. Performance is measured by KL divergence (estimated via kernel density estimation) between generated and true distributions, and computational cost via FLOPs and runtime (details in Appendix E.2).

Table 2: Comparison on KL divergence and FLOPs at different time steps. $H_{\text{pred}}$ indicates the maximum single-step prediction horizon for NSF; see text for details on recursive application. Average runtime per 100 samples for latent SDE (JAX): 124–148 ms; NSF (JAX): 0.3 ms (see Appendix E.2.5).

| Method | $t = 0.25$ | | $t = 0.5$ | | $t = 0.75$ | | $t = 1.0$ | |
|---|---|---|---|---|---|---|---|---|
| | KL | kFLOPs | KL | kFLOPs | KL | kFLOPs | KL | kFLOPs |
| Latent SDE [33] | $2.1 \pm 0.9$ | 959 | $1.8 \pm 0.1$ | 1,917 | $0.9 \pm 0.3$ | 2,839 | $1.5 \pm 0.5$ | 3,760 |
| Neural LSDE [38] | $1.3 \pm 0.4$ | 1,712 | $7.2 \pm 1.4$ | 3,416 | $74.5 \pm 24.6$ | 5,057 | $53.1 \pm 29.3$ | 6,699 |
| Neural GSDE [38] | $1.2 \pm 0.4$ | 1,925 | $3.9 \pm 0.3$ | 3,848 | $20.2 \pm 7.6$ | 5,698 | $14.1 \pm 8.4$ | 7,548 |
| Neural LNSDE [38] | $1.7 \pm 0.4$ | 1,925 | $4.6 \pm 0.7$ | 3,848 | $57.3 \pm 16.4$ | 5,698 | $44.6 \pm 23.4$ | 7,548 |
| SDE matching [3] | | | | | | | | |
| $\quad \Delta t = 0.0001$ | $4.3 \pm 0.7$ | 184,394 | $5.3 \pm 1.0$ | 368,787 | $3.4 \pm 0.8$ | 553,034 | $3.8 \pm 1.0$ | 737,354 |
| $\quad \Delta t = 0.01$ | $6.3 \pm 0.4$ | 1,917 | $11.7 \pm 0.5$ | 3,834 | $7.9 \pm 0.3$ | 5,677 | $6.0 \pm 0.3$ | 7,520 |
| NSF (ours) | | | | | | | | |
| $\quad H_{\text{pred}} = 1.0$ | $0.8 \pm 0.7$ | 53 | $1.3 \pm 0.1$ | 53 | $0.6 \pm 0.3$ | 53 | $0.2 \pm 0.6$ | 53 |
| $\quad H_{\text{pred}} = 0.5$ | $2.4 \pm 1.9$ | 53 | $1.3 \pm 0.1$ | 53 | $1.0 \pm 0.4$ | 105 | $1.7 \pm 1.1$ | 105 |
| $\quad H_{\text{pred}} = 0.25$ | $1.2 \pm 0.7$ | 53 | $1.2 \pm 0.1$ | 105 | $0.8 \pm 0.4$ | 156 | $1.3 \pm 0.8$ | 208 |

Fig. 3 shows paths from each method; NSF traces visually coincide with the true attractor whereas solver-based methods overspread. Quantitatively in Table 2, NSF ($H_{\text{pred}} = 1.0$, single-step) achieves the lowest KL divergence and with significantly fewer FLOPs. This leads to substantial runtime savings (approx. 0.3 ms vs. > 100 ms per batch, Appendix E.2.5). Notably, our flow-based training objective enables both single-step and recursive application variants to maintain relatively low KL divergence across different time horizons, with NSF remaining computationally cheaper in these configurations. This confirms NSF's capability to accurately model complex SDEs efficiently.

## 6.2 CMU Motion Capture

We evaluate Latent NSF on the CMU Motion Capture walking dataset [15]. The 23-sequence walking subset is down-sampled to 300 time steps and split 16/3/4 for train/validation/test, matching prior work [51, 54]. We report two established protocols.

Table 3: Test MSE and 95% confidence interval based on t-statistic on Motion Capture datasets. [†] indicates results from Yildiz et al. [54], [*] from Li et al. [33], [‡] from Course and Nair [9], and [§] from Ansari et al. [2]. All other results are our reproductions. Average runtime per 100 samples: latent SDE (JAX) - 75ms; Latent NSF (JAX) - 3.5ms (detailed in Appendix E.3.7).

| Methods | Setup 1 | Setup 2 |
|---|---|---|
| npODE [17] | $22.96^{†}$ | – |
| Neural ODE [6] | $22.49 \pm 0.88^{†}$ | – |
| ODE2VAE-KL [54] | $8.09 \pm 1.95^{†}$ | – |
| Latent ODE [43] | $5.98 \pm 0.28^{*}$ | $31.62 \pm 0.05^{§}$ |
| Latent SDE [33] | $12.91 \pm 2.90^{1}$ | $9.52 \pm 0.21^{§}$ |
| Latent Approx SDE [46] | $7.55 \pm 0.05^{§}$ | $10.50 \pm 0.86$ |
| ARCTA [9] | $7.62 \pm 0.93^{‡}$ | $9.92 \pm 1.82$ |
| NCDSSM [2] | $5.69 \pm 0.01^{§}$ | $4.74 \pm 0.01^{§}$ |
| SDE matching [3] | $\mathbf{5.20 \pm 0.43}^{2}$ | $4.26 \pm 0.35$ |
| Latent NSF (ours) | $8.62 \pm 0.32$ | $\mathbf{3.41 \pm 0.27}$ |

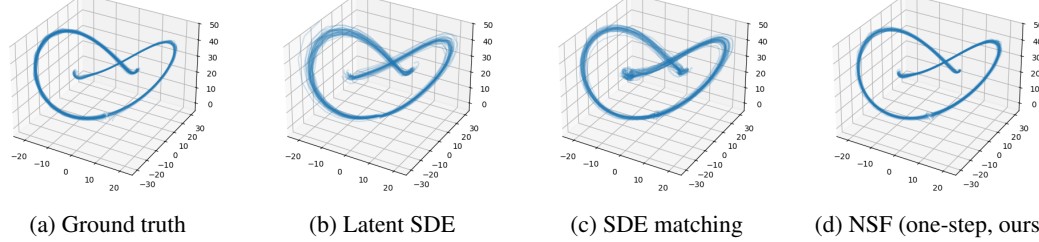

| (a) Ground truth | (b) Latent SDE | (c) SDE matching | (d) NSF (one-step, ours) |
|---|---|---|---|

Figure 3: Comparison of generated samples (64 samples per panel) on the stochastic Lorenz attractor. Baseline methods are simulated step-by-step. For NSF, each point is an independent sample from the learnt conditional distribution $p(\boldsymbol{x}_t \mid \boldsymbol{x}_s)$ originating from the same initial state and seed, connected visually for comparison. This visualisation choice aids in assessing distributional accuracy over time.

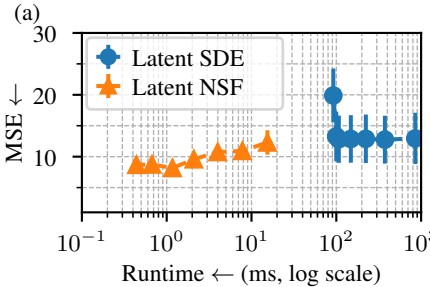 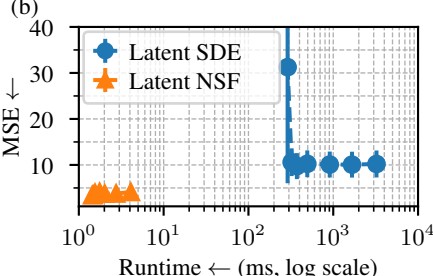

Figure 4: Trade-off between prediction accuracy (MSE) and computational cost (runtime, using JAX) for latent SDE and Latent NSF on CMU Motion Capture dataset. (a) Setup 1: Within-horizon forecasting. (b) Setup 2: Beyond-horizon extrapolation.

*Within-horizon forecasting (Setup 1)* [33, 54]. All 300 time steps are used during training; for tests, only the first three observations are revealed and the model must forecast the remaining 297 steps.

*Beyond-horizon extrapolation (Setup 2)* [2]. The last third segment of every sequence (steps 200–299) is withheld from training. At test time the model receives the first 100 observations and must predict the next 200 steps that lie beyond the training horizon.

Table 3 shows the results. In Setup 1, Latent NSF (single-step prediction) performs similarly to latent SDE models. Crucially, in the Setup 2 extrapolation task, latent NSF achieves state-of-the-art MSE.

These extrapolation gains align with the relevance of state-dependent stochasticity in human motion. Nonlinear SDE formulations often require learning complex, time-varying posterior structures (e.g., controlled SDEs or complicated conditional marginals). By contrast, Latent NSF directly models the Markov transition with conditional normalising flows, sidesteps these optimisation challenges.

Fig. 4 illustrates the trade-off between prediction accuracy and computational cost. While latent SDE adjusts the discretisation step size to balance accuracy and speed, Latent NSF controls this trade-off by varying the number of recursive applications. Latent NSF achieves better performance while being approximately more than two orders of magnitude faster than latent SDE in both setups, as detailed in Appendix E.3.7.

## 6.3 Stochastic Moving MNIST

Finally, we test Latent NSF on high-dimensional video using a Stochastic Moving MNIST [11] variant with physically plausible bouncing dynamics: digits undergo perfect specular reflection with small angular noise, avoiding the unphysical velocity resets of the original code. The task involves modelling two digits moving in a 64x64 frame. We train on 60k sequences and test on 10k held-out sequences. We evaluate using Fréchet distances (FD) on embeddings from a pre-trained SRVP model [14], measuring static content, dynamics, and frame similarity (validated in Appendix G). We compare our Latent NSF model against the latent SDE [33] as the main baseline, representing the current representative solver-based neural SDE model.

Table 4 and Fig. 5 show that Latent NSF (recursive and one-step) achieves competitive Static FD and Frame-wise FD compared to the latent SDE baseline. Dynamics FD for recursive NSF is slightly higher in this run, but visualisations suggest comparable dynamics capture (see Appendix E.4.5). The decline after step 35 across all models reflects the digits approaching a uniform spatial distribution, indicating a limitation of this metric when evaluating long-term predictions in feature spaces fitted to Gaussian distributions. Nevertheless, Latent NSF offers significant potential for computational speedup over the solver-based latent SDE (e.g., latent SDE: 751 ms vs. Latent NSF (one-step): 358

---

[1]Li et al. [33] report $4.03 \pm 0.20$ for Setup 1; our re-implementation and other reproductions observe higher MSE. Details are provided in Appendix E.3.2.

[2]Bartosh et al. [3] report $4.50 \pm 0.32$ for Setup 1; however, at the time of writing, official code for CMU Motion Capture is not yet available. We report our re-implementation here to maintain consistency across Setups 1 and 2. Details of our reproduction are provided in Appendix E.3.3.

Table 4: Fréchet distance comparison on Stochastic Moving MNIST using pre-trained SRVP embeddings. The lower the better. Note that dynamics FD is ill-defined for one-step simulation of latent NSF, which only predicts the marginal distribution of the future frames conditioned on the last observation.

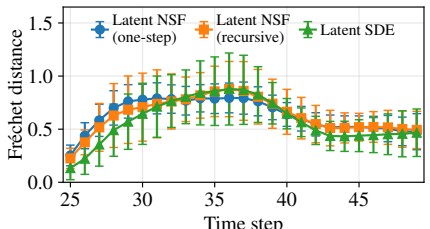

Figure 5: Comparison of frame-wise Fréchet distance across time-steps on Stochastic Moving MNIST.

| Methods | Static FD | Dynamics FD | Frame-wise FD |
|---|---|---|---|
| Latent SDE | $2.66 \pm 0.87$ | $5.39 \pm 3.10$ | $0.58 \pm 0.21$ |
| Latent NSF (recursive) | $2.36 \pm 0.60$ | $7.76 \pm 2.56$ | $0.63 \pm 0.17$ |
| Latent NSF (one-step) | $1.67 \pm 0.47$ | – | $0.63 \pm 0.15$ |

ms for full-sequence prediction; see Appendix E.4.5). This demonstrates NSF's promise for scaling to high-dimensional stochastic sequence modelling.

## 7 Discussion

*Neural Stochastic Flows* (NSFs) represent a novel paradigm for continuous-time stochastic modelling by directly learning weak solutions to SDEs as conditional distributions. Through carefully designed architectural constraints and regularisation objectives, NSF circumvents numerical integration while providing closed-form transition densities via normalising flows. The latent NSF extension enables applications to partially observed or high-dimensional time series, preserving efficient one-step transitions within a principled variational state-space modelling framework.

Our comprehensive evaluation across chaotic dynamical systems, human motion capture, and video sequences demonstrates that the proposed flow-based approach achieves comparable or superior performance relative to solver-based neural SDEs while reducing computational requirements by two orders of magnitude. Moreover, it attains state-of-the-art long-horizon extrapolation performance in latent settings. These results collectively establish that learning distributional flows, rather than the underlying vector fields required by numerical solvers, constitutes an effective and computationally efficient paradigm for real-time stochastic modelling.

**Limitations.** Several limitations remain. First, Chapman–Kolmogorov consistency is enforced only approximately via variational bounds, potentially yielding discrepancies beyond the training regime. We recommend systematic monitoring of flow losses on held-out data with appropriate adjustments. Second, the selection of maximum one-shot horizon necessitates careful balance between computational efficiency and predictive accuracy. Third, while our affine coupling architecture guarantees analytical Jacobian computation, it imposes constraints on the form of the architecture.

**Future work.** Extending NSF to action-conditioned settings could enable direct integration with control algorithms such as model predictive control and reinforcement learning. Additionally, bridging NSF with diffusion models presents opportunities for leveraging complementary strengths of both frameworks in continuous-time modelling. Another natural direction is to explore stronger flow parameterisations. In particular, transformer-based flows such as TarFlow [55] may relax the structural constraints of affine coupling while retaining tractable Jacobians, potentially further improving expressivity. These extensions would significantly broaden the applicability of NSF to decision-making and generative tasks.

Application-wise, the resulting analytical tractability, coupled with empirically demonstrated two-order-of-magnitude computational speedups, renders NSF particularly suitable for diverse applications, from high-frequency trading scenarios requiring real-time robotic control and fast simulations to large-scale digital twin implementations.

**Broader Impact Statement.** This paper presents work whose goal is to advance machine learning research. There may exist potential societal consequences of our work, however, none of which we feel must be specifically highlighted here at the moment of paper publication.

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

# Appendix Contents

# A  Architecture

## A.1  Neural Stochastic Flow

We refer to the main text for the NSF architecture and notation (Section 3, Eqs. (5) and (6) in the main text). Here we only note implementation specifics. $\mathrm{MLP}_\mu$ and $\mathrm{MLP}_\sigma$ are neural networks that share parameters and produce split outputs for the mean and variance components. Softplus activation is used for the variance component to ensure positivity. For stability, the scale heads end with a `tanh` multiplied by a learnt scalar to bound log-scales [13]. Exact log-densities are computed via the change-of-variables formula; see Section D.

## A.2  Bridge Model

Here, we describe the architecture of the bridge model used as auxiliary variational distribution [1, 40, 45]. The bridge model characterises the conditional distribution $b_{\boldsymbol\xi}(\boldsymbol{x}_t \mid \boldsymbol{x}_{t_i}, \boldsymbol{x}_{t_j})$ for any intermediate time point $t$ where $t_i < t < t_j$, given the boundary states. Combined with flow loss (Eq. (10)) minimisation, this enables inference of intermediate states when both endpoints are known. Let $\boldsymbol{c}_{\mathrm{br}} := (\boldsymbol{x}_{t_i}, \boldsymbol{x}_{t_j}, \Delta t, \tau, t_i)$ denote the conditioning parameters, where $\tau := (t - t_i)/\Delta t$ is the normalised time and $\Delta t := t_j - t_i$ is the time interval, and the initial time $t_i$ is ommitted for autonomous systems.

Similarly to the main NSF, we first draw a sample from a parametric Gaussian distribution:

$$\boldsymbol{z}_t = \underbrace{\boldsymbol{x}_{t_i} + \tau\big(\boldsymbol{x}_{t_j} - \boldsymbol{x}_{t_i}\big) + \alpha(\tau)\,\mathrm{MLP}_\mu(\boldsymbol{c}_{\mathrm{br}})}_{\boldsymbol{\mu}(t)} + \underbrace{\sqrt{\alpha(\tau)\Delta t}\,\mathrm{Softplus}\big(\mathrm{MLP}_\sigma(\boldsymbol{c}_{\mathrm{br}})\big)}_{\boldsymbol{\sigma}(t)} \odot \boldsymbol{\varepsilon}, \qquad (19)$$

where $\boldsymbol{\varepsilon} \sim \mathcal{N}(\boldsymbol{0}, \boldsymbol{I})$ is a standard normal noise vector, $\alpha(\tau) := \tau(1 - \tau)$ is the standard Brownian-bridge factor, and $\mathrm{MLP}_\mu$ and $\mathrm{MLP}_\sigma$ are neural networks that share parameters and produce split outputs for the mean and variance components. The intermediate state is then obtained by applying a flow transformation:

$$\boldsymbol{x}_t = \boldsymbol{f}_{\boldsymbol\xi}(\boldsymbol{z}_t, \boldsymbol{c}_{\mathrm{br}}) = \boldsymbol{f}_L(\cdot; \boldsymbol{c}_{\mathrm{br}}, \boldsymbol{\xi}_L) \circ \boldsymbol{f}_{L-1}(\cdot; \boldsymbol{c}_{\mathrm{br}}, \boldsymbol{\xi}_{L-1}) \circ \cdots \circ \boldsymbol{f}_1(\boldsymbol{z}_t; \boldsymbol{c}_{\mathrm{br}}, \boldsymbol{\xi}_1). \qquad (20)$$

Note that $\alpha(\tau)$ vanishes at the endpoints ($\tau = 0$ and $\tau = 1$, corresponding to $t = t_i$ and $t = t_j$), ensuring consistency with boundary conditions. The flow transformation $\boldsymbol{f}_{\boldsymbol\xi}$ is implemented using a series of conditioned bijective transformations, as detailed below.

### A.2.1  Conditioned Bijective Transformations

Each layer of the bridge flow follows the same coupling structure as in the main NSF (Section 3). Given an input $\boldsymbol{z}_t$, we split it into two parts $(\boldsymbol{z}_\mathrm{A}, \boldsymbol{z}_\mathrm{B}) = \mathrm{Split}(\boldsymbol{z}_t)$ with alternating partitioning across layers, and apply the transformation:

$$\begin{aligned}
&\boldsymbol{f}_i(\boldsymbol{z}_t; \boldsymbol{c}_{\mathrm{br}}, \boldsymbol{\xi}_i) \\
&= \mathrm{Concat}\Big(\boldsymbol{z}_\mathrm{A},\ \boldsymbol{z}_\mathrm{B} \odot \exp\big(\alpha(\tau)\,\mathrm{MLP}^{(i)}_{\mathrm{scale}}(\boldsymbol{z}_\mathrm{A}, \boldsymbol{c}_{\mathrm{br}}, \boldsymbol{\xi}^{(i)}_{\mathrm{scale}})\big) + \alpha(\tau)\,\mathrm{MLP}^{(i)}_{\mathrm{shift}}(\boldsymbol{z}_\mathrm{A}, \boldsymbol{c}_{\mathrm{br}}, \boldsymbol{\xi}^{(i)}_{\mathrm{shift}})\Big).
\end{aligned} \qquad (21)$$

Following the main model, we apply a hyperbolic tangent function with learnt scale as the final activation of $\mathrm{MLP}^{(i)}_{\mathrm{scale}}$ for training stability [13]. This construction preserves differentiability in $t$ and guarantees the consistency at the boundary conditions ($\tau = 0, 1$) where $\alpha(\tau) = 0$.

# B  Derivation of the Flow Loss

In our research, we utilise a bridge distribution described in Appendix A.2 as an auxiliary variational distribution [1, 40, 45] to constrain the upper bounds of bidirectional KL divergences between the one-step and two-step distributions of the NSF. By minimising these upper bounds, we guide the model towards satisfying the Chapman–Kolmogorov relation, as shown in Equation (4), thereby improving its consistency with the theoretical requirements of stochastic flows of diffeomorphisms combined with architecture design. Specifically, we define $\mathcal{L}_{\mathrm{flow}}$ as the expectation of a weighted sum of two components:

$$\mathcal{L}_{\mathrm{flow}}(\boldsymbol\theta, \boldsymbol\xi) = \mathbb{E}_{p(t_i, t_j, t_k)}\big[\lambda_{\text{1-to-2}}\mathcal{L}_{\text{flow, 1-to-2}}(\boldsymbol\theta, \boldsymbol\xi; t_i, t_j, t_k) + \lambda_{\text{2-to-1}}\mathcal{L}_{\text{flow, 2-to-1}}(\boldsymbol\theta, \boldsymbol\xi; t_i, t_j, t_k)\big]. \quad (22)$$

Here, $\lambda_{\text{1-to-2}}$ and $\lambda_{\text{2-to-1}}$ are weighting factors, which are set to 1 in the main text, balancing the contribution of each component to the overall flow loss. The two components, $\mathcal{L}_{\text{flow, 1-to-2}}$ and $\mathcal{L}_{\text{flow, 2-to-1}}$, correspond to the upper bounds of the one-step to two-step and two-step to one-step KL divergences, respectively.

In our experiment, which focuses on the autonomous case, the triplet $(t_i, t_j, t_k)$ is sampled according to the following procedure:

- The initial time $t_i$ is sampled from the data $\mathcal{D}$, representing the starting times in the dataset. (For non-autonomous SDEs, it is recommended to sample $t_i$ uniformly to ensure the Chapman–Kolmogorov equation is satisfied across all time points.)

- The final time $t_k$ is sampled from a mixture distribution $p(t_k) = \frac{1}{2}\mathcal{U}(t_i, t_i + H_{\text{train}}) + \frac{1}{2}p_{\text{data}}(t_k \mid t_i)$, where $\mathcal{U}(t_i, t_i + H_{\text{train}})$ denotes the uniform distribution over the interval $[t_i, t_i + H_{\text{train}}]$, and $p_{\text{data}}(t_k \mid t_i)$ is the conditional data distribution of end times corresponding to the sampled initial time $t_i$. Here, $H_{\text{train}}$ represents the maximum one-shot interval used during training.

- The intermediate time $t_j$ is uniformly sampled from the interval $[t_i, t_k]$, i.e., $t_j \sim \mathcal{U}(t_i, t_k)$.

The inclusion of $p_{\text{data}}(t_k \mid t_i)$ in the mixture distribution for sampling $t_k$ is motivated by our observation that the model achieves higher accuracy in regions where data exists. By incorporating this data-driven component, we aim to enhance the efficiency of the learning process, leveraging the model's improved performance in data-rich areas.

In the following, we derive the upper bounds of the one-step to two-step and two-step to one-step KL divergences.

## B.1 One-Step to Two-Step KL Divergence

We begin with the derivation of the upper bound of the KL divergence from a one-step computation to a marginalised two-step computation. To clarify the computation, subscripts $p_{\boldsymbol{\theta}}^1$ and $p_{\boldsymbol{\theta}}^2$ distinguish between the distributions for 1-step sampling from $\boldsymbol{x}_{t_i}$ to $\boldsymbol{x}_{t_k}$ and two-step sampling from $\boldsymbol{x}_{t_i}$ to $\boldsymbol{x}_{t_k}$ via $\boldsymbol{x}_{t_j}$, and the time arguments are omitted from the probability distributions for clarity. The key steps are outlined as follows:

$$D_{\text{KL}}\left(p_{\boldsymbol{\theta}}^1(\boldsymbol{x}_{t_k} \mid \boldsymbol{x}_{t_i}) \,\Big\|\, \int p_{\boldsymbol{\theta}}^2(\boldsymbol{x}_{t_k} \mid \boldsymbol{x}_{t_j}) \, p_{\boldsymbol{\theta}}^2(\boldsymbol{x}_{t_j} \mid \boldsymbol{x}_{t_i}) \, \mathrm{d}\boldsymbol{x}_{t_j}\right) \tag{23}$$

$$= \int p_{\boldsymbol{\theta}}^1(\boldsymbol{x}_{t_k} \mid \boldsymbol{x}_{t_i}) \log\left[\frac{p_{\boldsymbol{\theta}}^1(\boldsymbol{x}_{t_k} \mid \boldsymbol{x}_{t_i})}{\int p_{\boldsymbol{\theta}}^2(\boldsymbol{x}_{t_k} \mid \boldsymbol{x}_{t_j}) \, p_{\boldsymbol{\theta}}^2(\boldsymbol{x}_{t_j} \mid \boldsymbol{x}_{t_i}) \, \mathrm{d}\boldsymbol{x}_{t_j}}\right] \mathrm{d}\boldsymbol{x}_{t_k} \tag{24}$$

$$= \int p_{\boldsymbol{\theta}}^1(\boldsymbol{x}_{t_k} \mid \boldsymbol{x}_{t_i}) \log\left[\frac{p_{\boldsymbol{\theta}}^1(\boldsymbol{x}_{t_k} \mid \boldsymbol{x}_{t_i}) \, p_{\boldsymbol{\theta}}^2(\boldsymbol{x}_{t_j} \mid \boldsymbol{x}_{t_i}, \boldsymbol{x}_{t_k})}{p_{\boldsymbol{\theta}}^2(\boldsymbol{x}_{t_k} \mid \boldsymbol{x}_{t_j}) \, p_{\boldsymbol{\theta}}^2(\boldsymbol{x}_{t_j} \mid \boldsymbol{x}_{t_i})}\right] \mathrm{d}\boldsymbol{x}_{t_k} \tag{25}$$

$$= \iint p_{\boldsymbol{\theta}}^1(\boldsymbol{x}_{t_k} \mid \boldsymbol{x}_{t_i}) \, q_{\boldsymbol{\phi}}(\boldsymbol{x}_{t_j} \mid \boldsymbol{x}_{t_i}, \boldsymbol{x}_{t_k}) \left[\log\left[\frac{p_{\boldsymbol{\theta}}^1(\boldsymbol{x}_{t_k} \mid \boldsymbol{x}_{t_i}) \, q_{\boldsymbol{\phi}}(\boldsymbol{x}_{t_j} \mid \boldsymbol{x}_{t_i}, \boldsymbol{x}_{t_k})}{p_{\boldsymbol{\theta}}^2(\boldsymbol{x}_{t_k} \mid \boldsymbol{x}_{t_j}) \, p_{\boldsymbol{\theta}}^2(\boldsymbol{x}_{t_j} \mid \boldsymbol{x}_{t_i})}\right] \right.$$
$$\left. - \log\left[\frac{q_{\boldsymbol{\phi}}(\boldsymbol{x}_{t_j} \mid \boldsymbol{x}_{t_i}, \boldsymbol{x}_{t_k})}{p_{\boldsymbol{\theta}}^2(\boldsymbol{x}_{t_j} \mid \boldsymbol{x}_{t_i}, \boldsymbol{x}_{t_k})}\right]\right] \mathrm{d}\boldsymbol{x}_{t_j} \mathrm{d}\boldsymbol{x}_{t_k} \tag{26}$$

$$= \mathop{\mathbb{E}}_{p_{\boldsymbol{\theta}}^1(\boldsymbol{x}_{t_k} \mid \boldsymbol{x}_{t_i}) q_{\boldsymbol{\phi}}(\boldsymbol{x}_{t_j} \mid \boldsymbol{x}_{t_i}, \boldsymbol{x}_{t_k})}\left[\log\left[\frac{p_{\boldsymbol{\theta}}^1(\boldsymbol{x}_{t_k} \mid \boldsymbol{x}_{t_i}) \, q_{\boldsymbol{\phi}}(\boldsymbol{x}_{t_j} \mid \boldsymbol{x}_{t_i}, \boldsymbol{x}_{t_k})}{p_{\boldsymbol{\theta}}^2(\boldsymbol{x}_{t_k} \mid \boldsymbol{x}_{t_j}) \, p_{\boldsymbol{\theta}}^2(\boldsymbol{x}_{t_j} \mid \boldsymbol{x}_{t_i})}\right]\right.$$
$$\left. - D_{\text{KL}}\big(q_{\boldsymbol{\phi}}(\boldsymbol{x}_{t_j} \mid \boldsymbol{x}_{t_i}, \boldsymbol{x}_{t_k}) \,\|\, p_{\boldsymbol{\theta}}^2(\boldsymbol{x}_{t_j} \mid \boldsymbol{x}_{t_i}, \boldsymbol{x}_{t_k})\big)\right] \tag{27}$$

$$\leq \mathop{\mathbb{E}}_{p_{\boldsymbol{\theta}}^1(\boldsymbol{x}_{t_k} \mid \boldsymbol{x}_{t_i}) q_{\boldsymbol{\phi}}(\boldsymbol{x}_{t_j} \mid \boldsymbol{x}_{t_i}, \boldsymbol{x}_{t_k})}\left[\log\left[\frac{p_{\boldsymbol{\theta}}^1(\boldsymbol{x}_{t_k} \mid \boldsymbol{x}_{t_i}) \, q_{\boldsymbol{\phi}}(\boldsymbol{x}_{t_j} \mid \boldsymbol{x}_{t_i}, \boldsymbol{x}_{t_k})}{p_{\boldsymbol{\theta}}^2(\boldsymbol{x}_{t_k} \mid \boldsymbol{x}_{t_j}) \, p_{\boldsymbol{\theta}}^2(\boldsymbol{x}_{t_j} \mid \boldsymbol{x}_{t_i})}\right]\right] \tag{28}$$

$$=: \mathcal{L}_{\text{flow, 1-to-2}}(\boldsymbol{\theta}, \boldsymbol{\phi}; t_i, t_j, t_k). \tag{29}$$

The equation from the second line to the third line is obtained using Bayes' theorem and the Markov property:

$$\int p_{\boldsymbol{\theta}}^2(\boldsymbol{x}_{t_k} \mid \boldsymbol{x}_{t_j}) \, p_{\boldsymbol{\theta}}^2(\boldsymbol{x}_{t_j} \mid \boldsymbol{x}_{t_i}) \, \mathrm{d}\boldsymbol{x}_{t_j} = \frac{p_{\boldsymbol{\theta}}^2(\boldsymbol{x}_{t_k} \mid \boldsymbol{x}_{t_j}) \, p_{\boldsymbol{\theta}}^2(\boldsymbol{x}_{t_j} \mid \boldsymbol{x}_{t_i})}{p_{\boldsymbol{\theta}}^2(\boldsymbol{x}_{t_j} \mid \boldsymbol{x}_{t_i}, \boldsymbol{x}_{t_k})}. \tag{30}$$

### B.2 Two-Step to One-Step KL Divergence

Similarly, we now explore the derivation of the upper bound of the KL divergence from a marginalised two-step computation to a one-step computation. We denote this as $\mathcal{L}_{\text{flow, 2-to-1}}$. The key steps are outlined as follows:

$$D_{\mathrm{KL}}\left(\int p_{\boldsymbol{\theta}}^2(\boldsymbol{x}_{t_k} \mid \boldsymbol{x}_{t_j}) \, p_{\boldsymbol{\theta}}^2(\boldsymbol{x}_{t_j} \mid \boldsymbol{x}_{t_i}) \, \mathrm{d}\boldsymbol{x}_{t_j} \, \bigg\| \, p_{\boldsymbol{\theta}}^1(\boldsymbol{x}_{t_k} \mid \boldsymbol{x}_{t_i})\right) \tag{31}$$

$$= \int \left[\int p_{\boldsymbol{\theta}}^2(\boldsymbol{x}_{t_k} \mid \boldsymbol{x}_{t_j}) \, p_{\boldsymbol{\theta}}^2(\boldsymbol{x}_{t_j} \mid \boldsymbol{x}_{t_i}) \, \mathrm{d}\boldsymbol{x}_{t_j}\right] \log\left[\frac{p_{\boldsymbol{\theta}}^2(\boldsymbol{x}_{t_k} \mid \boldsymbol{x}_{t_j}) \, p_{\boldsymbol{\theta}}^2(\boldsymbol{x}_{t_j} \mid \boldsymbol{x}_{t_i})}{p_{\boldsymbol{\theta}}^2(\boldsymbol{x}_{t_j} \mid \boldsymbol{x}_{t_i}, \boldsymbol{x}_{t_k}) \, p_{\boldsymbol{\theta}}^1(\boldsymbol{x}_{t_k} \mid \boldsymbol{x}_{t_i})}\right] \mathrm{d}\boldsymbol{x}_{t_k} \tag{32}$$

$$= \iint p_{\boldsymbol{\theta}}^2(\boldsymbol{x}_{t_j} \mid \boldsymbol{x}_{t_i}) \, p_{\boldsymbol{\theta}}^2(\boldsymbol{x}_{t_k} \mid \boldsymbol{x}_{t_j}) \log\left[\frac{p_{\boldsymbol{\theta}}^2(\boldsymbol{x}_{t_j} \mid \boldsymbol{x}_{t_i}) \, p_{\boldsymbol{\theta}}^2(\boldsymbol{x}_{t_k} \mid \boldsymbol{x}_{t_j})}{p_{\boldsymbol{\theta}}^2(\boldsymbol{x}_{t_j} \mid \boldsymbol{x}_{t_i}, \boldsymbol{x}_{t_k}) \, p_{\boldsymbol{\theta}}^1(\boldsymbol{x}_{t_k} \mid \boldsymbol{x}_{t_i})}\right] \mathrm{d}\boldsymbol{x}_{t_j} \, \mathrm{d}\boldsymbol{x}_{t_k} \tag{33}$$

$$= \underset{p_{\boldsymbol{\theta}}^2(\boldsymbol{x}_{t_j} \mid \boldsymbol{x}_{t_i}) \, p_{\boldsymbol{\theta}}^2(\boldsymbol{x}_{t_k} \mid \boldsymbol{x}_{t_j})}{\mathbb{E}}\left[\log\left[\frac{p_{\boldsymbol{\theta}}^2(\boldsymbol{x}_{t_j} \mid \boldsymbol{x}_{t_i}) \, p_{\boldsymbol{\theta}}^2(\boldsymbol{x}_{t_k} \mid \boldsymbol{x}_{t_j})}{b_{\boldsymbol{\xi}}(\boldsymbol{x}_{t_j} \mid \boldsymbol{x}_{t_i}, \boldsymbol{x}_{t_k}) \, p_{\boldsymbol{\theta}}^1(\boldsymbol{x}_{t_k} \mid \boldsymbol{x}_{t_i})}\right]\right] \tag{34}$$

$$=: \mathcal{L}_{\text{flow, 2-to-1}}(\boldsymbol{\theta}, \boldsymbol{\xi}; t_i, t_j, t_k). \tag{35}$$

From the first line to the second line, we use the relationship in Equation (30). Although the expression inside the log in the second line appears to depend on $\boldsymbol{x}_{t_j}$, it does not actually depend on $\boldsymbol{x}_{t_j}$ due to this relationship. Therefore, the integrand change from the second line to the third line is justified.

By minimising flow loss consists of these upper bounds, we encourage the model to satisfy the flow property in both directions simultaneously.

## C  Detailed Description of the Learning Process

### C.1  Neural Stochastic Flow Optimisation

The optimisation process of NSF is summarised in Algorithm 1 in the main text. Here, we provide additional implementation details.

1. **Batch and time triplet sampling**: We sample a batch $\{(\boldsymbol{x}_{t_i}, t_i)_b\}_{b=1}^{B}$ from the dataset. For each element, we sample a triplet of time points $(t_i, t_j, t_k)$ where $t_i < t_j < t_k$, following the strategy described in Appendix B.

2. **Bridge model optimisation**: Before updating the main model parameters, we perform $K$ inner optimisation steps on the bridge model parameters $\boldsymbol{\xi}$. This ensures that the auxiliary variational distribution $b_{\boldsymbol{\xi}}(\boldsymbol{x}_{t_j} \mid \boldsymbol{x}_{t_i}, \boldsymbol{x}_{t_k})$ closely approximates the model's bridge distribution.

3. **Main model optimisation**: We then sample transition pairs $\{(\boldsymbol{x}_{t_i}, \boldsymbol{x}_{t_j}, t_i, t_j)_b\}_{b=1}^{B}$ from the dataset and compute the negative log-likelihood. Combined with the flow loss, this forms the total loss used to update the main model parameters $\boldsymbol{\theta}$.

This sequencing ensures that the main model is updated based on a bridge model that closely approximates the bridge distribution of the main model. Alternatively, the inner optimisation steps can be omitted and update the bridge model parameters simultaneously with the main model parameters when the main model parameters are updated.

### C.2 Latent Neural Stochastic Flow Optimisation

The optimisation process of Latent NSF is summarised in Algorithm 1 in the main text. Here, we provide additional implementation details.

1. **Observation encoding**: We sample batches of observation sequences $\{o_{t_{0:T},b}\}_{b=1}^{B}$ and encode them to latent states using the encoder network $q_\phi(x_{t_{0:T},b}|o_{t_{0:T},b})$ as defined in Eq. (15).

2. **Time triplet sampling**: Similar to NSF, we sample time triplets $(t_i, t_j, t_k)$ for each sequence, with $t_i < t_j < t_k$.

3. **Bridge model optimisation**: Before updating the main model parameters, we perform $K$ inner optimisation steps on the bridge model parameters $\xi$.

4. **Main model optimisation**: We then compute the total loss using Eq. (18) in the main text and update the NSF parameters $\theta$, encoder parameters $\phi$, and decoder parameters $\psi$.

The specific values of hyperparameters $\beta$, $\beta_{\text{skip}}$, and $\lambda$ in the total loss (Eq. (18) in the main text) vary depending on the dataset and task, and are detailed in Appendix E. As well as NSF, the inner optimisation steps can be omitted and update the auxiliary model parameters simultaneously with the main model parameters when the main model parameters are updated.

## D  Sampling and Density Evaluation

To generate a sample at time $t_j$ from a known state $x_{t_i}$ at time $t_i$, we compute $\mu(x_{t_i}, t_i, \Delta t)$ and $\sigma(x_{t_i}, t_i, \Delta t)$ using the MLPs, draw $\varepsilon \sim \mathcal{N}(0, I)$ and compute $z = \mu + \sigma \odot \varepsilon$. Then, apply the flow transformation: $x_{t_j} = f_L \circ f_{L-1} \circ \cdots \circ f_1(z; \Delta t, t_i)$. This enables single-step prediction without numerical integration, following the standard approach used in normalising flows.

Using the change of variables formula, the log-density can be computed as:

$$\log p_\theta(x_{t_j} \mid x_{t_i}; t_i, \Delta t) = \log p(\varepsilon) - \sum_{i=1}^{L} \log \left| \det \frac{\partial f_i}{\partial f_{i-1}} \right|, \tag{36}$$

where $p(\varepsilon)$ is the density of the standard normal distribution. When using the affine coupling formulation, this simplifies to:

$$\log p_\theta(x_{t_j} \mid x_{t_i}; t_i, \Delta t) = -\frac{1}{2}\|\varepsilon\|^2 - \frac{d}{2}\log 2\pi - \sum_{i=1}^{L} \sum_{j \in B_i} \Delta t \cdot \text{MLP}_{\text{scale}}^{(i)}(z_A^{(i-1)}, t_i, \Delta t)_j, \tag{37}$$

where $B_i$ is the set of indices corresponding to the second partition in the $i$-th coupling layer, and $z^{(i-1)}$ is the output of the $(i-1)$-th layer. This density evaluation approach mirrors the standard technique in normalising flows, with appropriate conditioning on the time parameters.

## E  Experimental Details

This section provides detailed information about the experimental setups, hyperparameters, architectures, and computational analyses for the experiments presented in Section 6 in the main text. Implementations are available at `https://github.com/nkiyohara/jax_nsf`.

### E.1  General Setup

Unless otherwise specified, experiments were conducted with the following setup:

- Frameworks:
    - Our (Latent) NSF models: JAX [5] with Equinox library [24]
    - PyTorch baselines: torchsde [33]
    - JAX baselines: Diffrax [23]
- Hardware: NVIDIA RTX 3090 GPU
- Optimiser: AdamW [27, 37]
- Hyperparameters: Specified below for each experiment

### E.2 Stochastic Lorenz Attractor

#### E.2.1 Data Generation

Following Li et al. [33], the data generation process is as follows:

- SDE:
$$\mathrm{d}x = \sigma(y - x)\,\mathrm{d}t + \alpha_x\,\mathrm{d}W_1$$
$$\mathrm{d}y = (x(\rho - z) - y)\,\mathrm{d}t + \alpha_y\,\mathrm{d}W_2$$
$$\mathrm{d}z = (xy - \beta z)\,\mathrm{d}t + \alpha_z\,\mathrm{d}W_3$$

- Parameters: $\sigma = 10, \rho = 28, \beta = 8/3, \alpha = (0.15, 0.15, 0.15)$
- Initial states $(x_0, y_0, z_0)$: Sampled from $\mathcal{N}(\mathbf{0}, \boldsymbol{I})$
- Trajectories: 1,024 for training and 1,024 for testing
- Time steps: $t \in [0, 1]$ with time step size $\Delta t = 0.025$ (41 time steps per trajectory)

#### E.2.2 Configurations for Neural Stochastic Flows

- Architecture:
    - State dimension `d_state` = 3; conditioning dimension `d_cond` = 4 (state + time)
    - Gaussian parameter network: `Input(d_cond)` $\to$ 2$\times$`[Linear(64)` $\to$ `SiLU()]` $\to$ `Linear(2*d_state)`; splits into (`mean`, `std`) with std via `Softplus()`
    - Scale-shift networks (in conditional flow): `Input(d_state/2 + d_cond)` $\to$ 2$\times$`[Linear(64)` $\to$ `SiLU()]` $\to$ `Linear(d_state)`; splits into (`scale`, `shift` in Eq. (7) in the main text) (In this case, since `d_state` is odd, we split three dimensions of `d_state` into one and two dimensions)
    - Conditional flow (affine coupling; Eq. (5), (6)): 4 layers with alternating masking
- Parameters:
    - NSF: 25,130
    - Bridge model: 26,410
- Training:
    - Data conversion: Time series data is converted into pairs of states $(x_{t_i}, x_{t_j})$ where $t_j > t_i$, to predict $p(x_{t_j} \mid x_{t_i})$
    - Epochs: 1000
    - Batch size: 256
    - Optimiser: AdamW [27, 37]
    - Learning rate: 0.001
    - Weight decay: $10^{-5}$
    - Loss function: Combined negative log-likelihood and flow loss (Eq. (11))
        * $\lambda = 0.4$ (0.2 for data component, 0.2 for sampled component)
        * $\lambda_{\text{1-to-2}} = \lambda_{\text{2-to-1}} = 1.0$
    - Time triplets $(t_i, t_j, t_k)$ for $\mathcal{L}_{\text{flow}}$: Sampled as described in Appendix B with $H_{\text{train}} = 1$
    - Auxiliary updates: bridge model $b_{\boldsymbol{\xi}}$ trained concurrently with $K = 5$ inner optimisation steps per main model update

#### E.2.3 Configurations for Baselines

For our baseline comparisons, we utilised the official implementations of latent SDE [33], SDE matching [3], SDE-GAN [25], and Stable Neural SDE series [38] with minimal modifications. We maintained the original model architectures, optimiser configurations, and hyperparameter settings as provided in their respective codebases, replacing only the input data to match our experimental setting. For SDE-GAN, we found that it exhibited significant training instability when applied to the Stochastic Lorenz Attractor dataset, preventing convergence despite multiple training attempts with varying hyperparameter configurations. Therefore, we excluded SDE-GAN results from our comparison in the paper.

**Latent SDE.**  The latent SDE model [33] has the following configuration:

- Architecture:
  - Observation dimension `d_obs = 3`; latent dimension `d_latent = 4`; hidden size `d_hidden = 128`; sequence length `T = 41`
  - Encoder: `Input((T, d_obs))` $\xrightarrow[\text{reverse in time}]{}$ `GRU(128)` $\rightarrow$ `Linear(d_context)` $\xrightarrow[\text{restore time order}]{}$ `Output((T, d_context))`
  - Posterior drift network: `Input(d_latent + d_context)` $\rightarrow$ `2×[Linear(128)` $\rightarrow$ `Softplus()]` $\rightarrow$ `Linear(d_latent)`
  - Prior drift network: `Input(d_latent)` $\rightarrow$ `2×[Linear(128)` $\rightarrow$ `Softplus()]` $\rightarrow$ `Linear(d_latent)`
  - Diffusion networks: per-dimension `Input(1)` $\rightarrow$ `Linear(128)` $\rightarrow$ `Softplus()` $\rightarrow$ `Linear(1)` $\rightarrow$ `Sigmoid()`
  - Decoder: `Input(d_latent)` $\rightarrow$ `Linear(d_obs)`
- Parameters:
  - Encoder: 59,328
  - Posterior initial state network: 520
  - Posterior drift network: 25,860
  - Prior drift network: 17,668
  - Diffusion networks: 1,540
  - Decoder: 15
- Training:
  - Iterations: 5000
  - Optimiser: Adam
  - Initial learning rate: 0.01
  - Learning rate decay: Exponential (gamma=0.997)
  - KL divergence annealing: Linear annealing from 0 to 1 over first 1000 iterations
  - Numerical integration: Euler–Maruyama method with step size $\Delta t = 0.01$

**SDE Matching.**  The SDE matching model [3] has the following configuration:

- Architecture:
  - Observation dimension `d_obs = 3`; latent dimension `d_latent = 4`; hidden size `d_hidden = 128`
  - Encoder: `Input((T, d_obs))` $\rightarrow$ `GRU(d_hidden)` $\rightarrow$ `Output((T+1, d_hidden))` (concatenate initial hidden with per-step outputs)
  - Prior initial distribution: learnable diagonal Gaussian; parameters `m, log s` in `d_latent` dimensions each
  - Prior drift network: `Input(d_latent)` $\rightarrow$ `Linear(d_hidden)` $\rightarrow$ `Softplus()` $\rightarrow$ `Linear(d_hidden)` $\rightarrow$ `Softplus()` $\rightarrow$ `Linear(d_latent)`
  - Diffusion networks: per-dimension `Input(1)` $\rightarrow$ `Linear(d_hidden)` $\rightarrow$ `Softplus()` $\rightarrow$ `Linear(1)` $\rightarrow$ `Sigmoid()`
  - Posterior affine head: `Input(d_hidden + 1)` $\rightarrow$ `Linear(d_hidden)` $\rightarrow$ `SiLU()` $\rightarrow$ `Linear(d_hidden)` $\rightarrow$ `SiLU()` $\rightarrow$ `Linear(2*d_latent)`; splits into $(\mu_t, \log \sigma_t)$ with $\sigma_t = \exp(\log \sigma_t)$
  - Decoder: `Input(d_latent)` $\rightarrow$ `Linear(d_obs)`; Gaussian likelihood with fixed std $\sigma = 0.01$
- Parameters:
  - Prior init: 8
  - Prior drift network: 17,668
  - Diffusion networks: 1,540

- Decoder: 15
- Encoder: 99,072
- Posterior affine head: 66,560

- Training:

  - Iterations: 10,000
  - Batch size: 1,024
  - Optimiser: Adam
  - Learning rate: 0.001

**Stable Neural SDE Series.** The Stable Neural SDE series includes Neural LSDE, Neural GSDE, and Neural LNSDE.

- Architecture (common to all variants unless specified):

  - Encoder: `Input(3 + 1)` $\rightarrow$ `Linear(64)`
  - Embedding network: `Input(64)` $\rightarrow$ `Linear(64)`
  - Drift network: `Input(64)` $\rightarrow$ $2\times$`[Linear(64)` $\rightarrow$ `LipSwish()]` $\rightarrow$ `Linear(64)`
  - Diffusion network: `Input(64)` $\rightarrow$ $2\times$`[Linear(64)` $\rightarrow$ `LipSwish()]` $\rightarrow$ `Linear(64)`
  - Decoder: `Input(64)` $\rightarrow$ `Linear(3)`

- Parameters:

  - Encoder: 256
  - Embedding network: 4,160
  - Drift network: 12,480
  - Diffusion network: 12,480
  - Decoder: 192

- Training:

  - Epochs: 100
  - Batch size: 16
  - Optimiser: Adam
  - Learning rate: 0.001
  - Numerical integration: Euler–Maruyama scheme with step-size $\Delta t = 0.01$

### E.2.4 Evaluation and Metrics

**KL Divergence.** For computing the KL divergence, we employed non-parametric Kernel Density Estimation (KDE) with Gaussian kernels. The bandwidth parameter was determined through 3-fold cross-validation by optimising the log-likelihood on unseen data. To ensure statistical robustness, we computed the KL divergence and test log-likelihood across ten independent folds, excluding values beyond the 5th and 95th percentiles to mitigate outlier influence. The final reported metrics consist of the mean and standard deviation of these filtered values.

**FLOPs.** We calculated FLOPs on a per-sample basis using different methods depending on the framework. For JAX-based models like Neural Stochastic Flows, we analysed the JIT-compiled sampling function using `jax.jit().lower().compile().cost_analysis()['flops']`. For PyTorch-based models such as latent SDE and Stable Neural SDEs, we utilised `torch.utils.flop_counter.FlopCounterMode` to count FLOPs during sample generation, then divided by batch size. Our comparison table shows the FLOPs value for one sample.

**Runtime.** For all methods, we measured the wall-clock runtime for computing vectorised batches of 100 samples.

### E.2.5 Detailed Results

Table 6 presents FLOPs comparisons across different prediction horizons. NSF requires constant FLOPs for single-step predictions regardless of time interval, while baseline methods scale linearly with the number of integration steps.

Table 7 shows wall-clock runtime measurements for generating 100 samples. PyTorch implementations exhibit higher runtime due to dynamic graph construction, while JAX implementations benefit from JIT compilation. NSF achieves consistent sub-millisecond runtime across all time horizons.

### E.3 CMU Motion Capture

#### E.3.1 Data Preparation

Data was prepared as follows:

- Dataset: Preprocessed CMU Motion Capture (walking) from Yildiz et al. [54]
- Features: 50-dimensional joint angle vectors
- Sequence length: 300 time steps
- Split: 16 training, 3 validation, and 4 test sequences
- Setups:
    - Setup 1 [33]: Train on full 300 steps, test by predicting steps 4–300 given steps 1–3
    - Setup 2 [2]: Train on steps 1–200, test by predicting steps 101–300 given steps 1–100

#### E.3.2 Latent SDE Reproduction

As noted in the footnote 1 in the main text, whilst Li et al. [33] report $4.03 \pm 0.20$ for Setup 1, our re-implementation and other reproductions observe higher MSE values. These results are consistent with those reported in public discussions `https://github.com/google-research/torchsde/issues/112`. Our implementation is available at `https://github.com/nkiyohara/latent-sde-mocap`, which is consistent with the results reported in the public discussion.

#### E.3.3 SDE Matching Reproduction

As noted in the footnote 2 in the main text, at the time of writing, the official implementation of SDE matching [3] for the CMU Motion Capture dataset has not yet been publicly released. To ensure consistent comparison across Setup 1 and Setup 2, we re-implemented SDE matching to the best of our ability based on the description in their paper. While Bartosh et al. [3] report $4.50 \pm 0.32$ for Setup 2, our re-implementation achieves $5.20 \pm 0.43$. This discrepancy may stem from implementation details not specified in the original paper, such as architectural choices, hyperparameter settings, or training procedures. We have made our best efforts obtain the smaller MSE by adjusting the hyperparameters, architecture, and training procedure. Our implementation is available at `https://github.com/nkiyohara/sde-matching-mocap`.

#### E.3.4 Configurations for Latent NSF

**Latent NSF — Setup 1 (within-horizon forecasting).**

- Architecture:
    - Latent dimension `d_latent = 6`; conditioning dimension `d_cond = 7`
    - Encoder: `Input(150)` $\rightarrow$ `2×[Linear(30)` $\rightarrow$ `Softplus()]` $\rightarrow$ `Linear(2*d_latent)`; splits into (mean, std) with std via `Softplus()`
    - Emission $p_\psi(o_{t_i} \mid x_{t_i})$: `Input(d_latent)` $\rightarrow$ `2×[Linear(30)` $\rightarrow$ `Softplus()]` $\rightarrow$ `Linear(50)`; Gaussian likelihood with fixed std
    - NSF transition model (latent prior):
        * Gaussian parameter network: `Input(d_cond)` $\rightarrow$ `2×[Linear(30)` $\rightarrow$ `SiLU()]` $\rightarrow$ `Linear(2*d_latent)`; splits into (mean, std) with std via `Softplus()`

Table 5: FLOPs comparison (K) for Stochastic Lorenz Attractor experiments, showing computational costs per single sample prediction. EM refers to Euler–Maruyama method, and SRK refers to Stochastic Runge–Kutta method.

| Method | $t = 0.25$ | $t = 0.5$ | $t = 0.75$ | $t = 1.0$ |
|---|---|---|---|---|
| Latent SDE [33] | | | | |
| EM, $\Delta t = 0.003$ | 3,133 | 6,193 | 9,253 | 12,349 |
| EM, $\Delta t = 0.01$ | 959 | 1,917 | 2,839 | 3,760 |
| EM, $\Delta t = 0.03$ | 369 | 664 | 995 | 1,290 |
| Milstein, $\Delta t = 0.01$ | 1,010 | 2,021 | 2,994 | 3,967 |
| SRK, $\Delta t = 0.01$ | 9,253 | 18,838 | 28,054 | 37,270 |
| SDE matching [3] | | | | |
| EM, $\Delta t = 0.0001$ | 184,394 | 368,787 | 553,034 | 737,354 |
| EM, $\Delta t = 0.001$ | 18,506 | 37,011 | 55,443 | 73,875 |
| EM, $\Delta t = 0.01$ | 1,917 | 3,834 | 5,677 | 7,520 |
| Stable Neural SDEs [38] | | | | |
| Neural LSDE (EM, $\Delta t = 0.01$) | 1,708 | 3,416 | 5,057 | 6,699 |
| Neural LSDE (Milstein, $\Delta t = 0.01$) | 1,708 | 3,416 | 5,057 | 6,699 |
| Neural LSDE (SRK, $\Delta t = 0.01$) | 16,483 | 33,555 | 49,971 | 66,387 |
| Neural GSDE (EM, $\Delta t = 0.01$) | 1,925 | 3,848 | 5,698 | 7,548 |
| Neural GSDE (Milstein, $\Delta t = 0.01$) | 1,925 | 3,848 | 5,698 | 7,548 |
| Neural GSDE (SRK, $\Delta t = 0.01$) | 18,571 | 37,807 | 56,303 | 74,799 |
| Neural LNSDE (EM, $\Delta t = 0.01$) | 1,925 | 3,848 | 5,698 | 7,548 |
| Neural LNSDE (Milstein, $\Delta t = 0.01$) | 1,925 | 3,848 | 5,698 | 7,548 |
| Neural LNSDE (SRK, $\Delta t = 0.01$) | 18,571 | 37,807 | 56,303 | 74,799 |
| NSF (ours) | | | | |
| $H_{\text{train}} = 1.0, H_{\text{pred}} = 0.25$ | 53 | 105 | 156 | 208 |
| $H_{\text{train}} = 1.0, H_{\text{pred}} = 0.5$ | 53 | 53 | 105 | 105 |
| $H_{\text{train}} = 1.0, H_{\text{pred}} = 1.0$ | 53 | 53 | 53 | 53 |

Table 6: FLOPs comparison (K) for Stochastic Lorenz Attractor experiments, showing computational costs per single sample prediction. EM refers to Euler–Maruyama method, and SRK refers to Stochastic Runge–Kutta method.

| Method | $t = 0.25$ | $t = 0.5$ | $t = 0.75$ | $t = 1.0$ |
|---|---|---|---|---|
| Latent SDE (EM, $\Delta t = 0.01$) | 959 | 1,917 | 2,839 | 3,760 |
| Latent SDE (Milstein, $\Delta t = 0.01$) | 1,010 | 2,021 | 2,994 | 3,967 |
| Latent SDE (SRK, $\Delta t = 0.01$) | 9,253 | 18,838 | 28,054 | 37,270 |
| Neural LSDE (EM, $\Delta t = 0.01$) | 1,708 | 3,416 | 5,057 | 6,699 |
| Neural LSDE (Milstein, $\Delta t = 0.01$) | 1,708 | 3,416 | 5,057 | 6,699 |
| Neural LSDE (SRK, $\Delta t = 0.01$) | 16,483 | 33,555 | 49,971 | 66,387 |
| Neural GSDE (EM, $\Delta t = 0.01$) | 1,925 | 3,848 | 5,698 | 7,548 |
| Neural GSDE (Milstein, $\Delta t = 0.01$) | 1,925 | 3,848 | 5,698 | 7,548 |
| Neural GSDE (SRK, $\Delta t = 0.01$) | 18,571 | 37,807 | 56,303 | 74,799 |
| Neural LNSDE (EM, $\Delta t = 0.01$) | 1,925 | 3,848 | 5,698 | 7,548 |
| Neural LNSDE (Milstein, $\Delta t = 0.01$) | 1,925 | 3,848 | 5,698 | 7,548 |
| Neural LNSDE (SRK, $\Delta t = 0.01$) | 18,571 | 37,807 | 56,303 | 74,799 |
| NSF ($H_{\text{train}} = 0.25, H_{\text{pred}} = 0.25$) | 53 | 105 | 156 | 208 |
| NSF ($H_{\text{train}} = 0.5, H_{\text{pred}} = 0.5$) | 53 | 53 | 105 | 105 |
| NSF ($H_{\text{train}} = 1.0, H_{\text{pred}} = 1.0$) | 53 | 53 | 53 | 53 |

* Scale-shift network (in conditional flow): 4 layers with alternating masking, each layer: `Input(d_latent/2 + d_cond)` → `2×[Linear(30)` → `SiLU()]` → `Linear(d_latent)`; splits into (`scale`, `shift` in Eq. (7) in the main text)

  – Bridge model:
    * Gaussian parameter network: `Input(d_cond + 1)` → `2×[Linear(30)` → `SiLU()]` → `Linear(2*d_latent)`; splits into (`mean`, `std`) with std via `Softplus()`
    * Scale-shift network (in conditional flow): 4 layers with alternating masking, each layer: `Input(d_latent + d_cond + 1)` → `2×[Linear(30)` → `SiLU()]` → `Linear(d_latent)`; splits into (`scale`, `shift` in Eq. (20))

- **Parameters:**
  – Stochastic flow: 8,454
  – Bridge model: 9,504
  – Decoder: 2,690
  – Posterior: 5,832

- **Training:**
  – Objective: maximise Eq. (18)
  – Steps: 100,000
  – Batch size: 128
  – Optimiser: AdamW
  – Learning rate: 0.001
  – Hyperparameters: $\beta = 0.1$, $\lambda = 1.0$, $\beta_{\text{skip}} = 0.1$
  – Skip-ahead KL: prediction horizons up to 3 time steps; 10 samples per skip

## Latent NSF — Setup 2 (extrapolation).

- **Architecture:**
  – Latent dimension `d_latent = 6`, conditioning dimension `d_cond = 7`
  – Encoder: `Input((T, 50))` → `GRU(32)` → `2×[Linear(32)` → `Softplus()]` → `Linear(2*d_latent)`; splits into (`mean`, `std`) with std via `Softplus()`
  – Emission $p_\psi(\boldsymbol{o}_{t_i} \mid \boldsymbol{x}_{t_i})$: `Input(d_latent)` → `2×[Linear(30)` → `Softplus()]` → `Linear(50)`; Gaussian likelihood with trainable std
  – NSF transition model (latent prior):
    * Gaussian parameter network: `Input(d_cond)` → `2×[Linear(64)` → `SiLU()]` → `Linear(2*d_latent)`; splits into (`mean`, `std`) with std via `Softplus()`
    * Scale-shift network (in conditional flow): 4 layers with alternating masking, each layer: `Input(d_latent/2 + d_cond)` → `2×[Linear(64)` → `SiLU()]` → `Linear(d_latent)`; splits into (`scale`, `shift`)
  – Bridge model:
    * Gaussian parameter network: `Input(d_cond + 1)` → `2×[Linear(32)` → `SiLU()]` → `Linear(2*d_latent)`; splits into (`mean`, `std`) with std via `Softplus()`
    * Scale-shift network (in conditional flow): 4 layers with alternating masking, each layer: `Input(d_latent/2 + d_cond + 1)` → `2×[Linear(32)` → `SiLU()]` → `Linear(d_latent)`; splits into (`scale`, `shift`)

- **Parameters:**
  – Stochastic flow: 28,820
  – Bridge model: 10,452
  – Decoder: 4,240
  – Posterior: 10,604

- **Training:**
  – Objective: maximise Eq. (18)

- Steps: 100,000
- Batch size: 64
- Optimiser: AdamW
- Learning rate: 0.001
- Hyperparameters: $\beta = 0.3$, $\lambda = 1.0$, $\beta_{\text{skip}} = 0.3$
- Warm-up: $\beta$ and flow loss weight are linearly increased over the first 2,000 steps
- Time horizon for flow loss: $H_{\text{train}} = 20$
- Auxiliary updates: $K = 3$ inner steps per training iteration for the bridge model
- Skip-ahead KL: prediction horizons up to $\tau = 10$ time steps; 10 samples per skip

### E.3.5   Configurations for Baselines

**Latent SDE — Setup 1 (within-horizon forecasting).**

- Architecture:
  - Observation dimension `d_obs = 50`; latent dimension `d_latent = 6`; context dimension `d_context = 3`
  - Encoder: `Input(150)` $\rightarrow$ `2×[Linear(30)` $\rightarrow$ `Softplus()]` $\rightarrow$ `Linear(15)`; splits into (`mean`, `std`, `context`) with `std` via `exp()`, with 6, 6, and 3 dimensions respectively
  - Posterior drift network: `Input(d_latent + 1 + d_context)` $\rightarrow$ `Linear(30)` $\rightarrow$ `Softplus()` $\rightarrow$ `Linear(d_latent)`
  - Prior drift network: `Input(d_latent + 1)` $\rightarrow$ `Linear(30)` $\rightarrow$ `Softplus()` $\rightarrow$ `Linear(d_latent)`
  - Diffusion networks: per-latent `Input(2)` $\rightarrow$ `Linear(30)` $\rightarrow$ `Softplus()` $\rightarrow$ `Linear(1)` $\rightarrow$ `Sigmoid()`
  - Decoder: `Input(d_latent)` $\rightarrow$ `Linear(30)` $\rightarrow$ `Softplus()` $\rightarrow$ `Linear(30)` $\rightarrow$ `Softplus()` $\rightarrow$ `Linear(2*d_obs)`

- Parameters:
  - Encoder: 5,925
  - Decoder: 4,240
  - Posterior drift network: 516
  - Prior drift network: 426
  - Diffusion networks: 726

- Training:
  - Iterations: 5,000
  - Optimiser: Adam
  - Learning rate: 0.01 with exponential decay $\gamma = 0.999$ at every step
  - KL annealing: linear over the first 400 iterations
  - Numerical integration: Euler–Maruyama with step size $\Delta t = 0.02$

**Latent SDE — Setup 2 (extrapolation).**

- Architecture:
  - Observation dimension `d_obs = 50`; latent dimension `d_latent = 10`; context dimension `d_context = 3`
  - Encoder: `Input((T, 50))` $\rightarrow$ `ODE-GRU(30)`; linear heads: `Input(30)` $\rightarrow$ `Linear(3)` and `Input(30)` $\rightarrow$ `Linear(20)`; splits into (`mean`, `std`) with `std` via `exp()`
  - Posterior drift network: autonomous `Input(d_latent + d_context)` $\rightarrow$ `Linear(30)` $\rightarrow$ `Softplus()` $\rightarrow$ `Linear(d_latent)`
  - Prior drift network: `Input(d_latent)` $\rightarrow$ `Linear(30)` $\rightarrow$ `Softplus()` $\rightarrow$ `Linear(d_latent)`

- Diffusion networks: per-latent `Input(1)` $\rightarrow$ `Linear(30)` $\rightarrow$ `Softplus()` $\rightarrow$ `Linear(1)` $\rightarrow$ `Sigmoid()`
- Decoder: `Input(d_latent)` $\rightarrow$ `Linear(30)` $\rightarrow$ `Softplus()` $\rightarrow$ `Linear(30)` $\rightarrow$ `Softplus()` $\rightarrow$ `Linear(2*d_obs)`

- Parameters:
  - Encoder: 12,027
  - Decoder: 4,360
  - Posterior drift network: 730
  - Prior drift network: 640
  - Diffusion networks: 910

- Training:
  - Iterations: 5,000
  - Optimiser: Adam
  - Learning rate: 0.01 with exponential decay $\gamma = 0.9$ every 500 iterations
  - KL annealing: Linear over the first 500 iterations
  - Numerical integration: Euler–Maruyama with step size $\Delta t = 0.05$

**SDE Matching – Setup 1 (within-horizon forecasting).**

- Architecture:
  - Observation dimension `d_obs = 50`; latent dimension `d_latent = 6`; posterior hidden size `d_hidden = 100`
  - Encoder: `Input((T, d_obs))` $\xrightarrow[\text{reverse in time}]{}$ `GRU(d_hidden)` $\xrightarrow[\text{restore time order}]{}$ `Output((T+1, d_hidden))` (concatenate initial hidden with per-step outputs)
  - Posterior affine head: `Input(d_hidden + 1)` $\rightarrow$ `Linear(d_hidden)` $\rightarrow$ `SiLU()` $\rightarrow$ `Linear(d_hidden)` $\rightarrow$ `SiLU()` $\rightarrow$ `Linear(2*d_latent)`; splits into $(\mu_t, \sigma_t)$ with $\sigma_t = $ `Softplus`$(\cdot)$; time-weighted aggregation over encoder outputs enabled
  - Prior initial distribution: learnable diagonal Gaussian; parameters `m`, `log s` in $\mathbb{R}^{d\_latent}$
  - Prior drift network (autonomous): `Input(d_latent)` $\rightarrow$ `Linear(30)` $\rightarrow$ `Softplus()` $\rightarrow$ `Linear(d_latent)`
  - Diffusion networks (diagonal, per-latent): `Input(1)` $\rightarrow$ `Linear(30)` $\rightarrow$ `Softplus()` $\rightarrow$ `Linear(1)` $\rightarrow$ `Sigmoid()`
  - Decoder (observation model): `Input(d_latent)` $\rightarrow$ $2\times$`[Linear(30)` $\rightarrow$ `Softplus()]` $\rightarrow$ `Linear(d_obs)`; Gaussian likelihood with fixed std ($\sigma_0 = 1.0$, floor $10^{-3}$)

- Parameters:
  - Prior SDE: 942
  - Prior Observation: 2,740
  - Posterior Encoder: 45,400
  - Posterior Affine: 21,513

- Training:
  - Iterations: 2,000
  - Optimiser: AdamW
  - Learning rate: 0.01 with exponential decay $\gamma = 0.999$ per step
  - KL weight: 1.0
  - Training sequence length: sampled from $\mathcal{U}\{3, 100\}$

**SDE Matching – Setup 2 (extrapolation).**

- Architecture:
  - Observation dimension `d_obs = 50`; latent dimension `d_latent = 10`; posterior hidden size `d_hidden = 1000`; prior SDE: autonomous

- Encoder: `Input((T, d_obs))` $\xrightarrow[\text{reverse in time}]{}$ `GRU(d_hidden)` $\xrightarrow[\text{restore time order}]{}$ `Output((T+1, d_hidden))`
- Posterior affine head: `Input(d_hidden)` $\rightarrow$ `Linear(d_hidden)` $\rightarrow$ `SiLU()` $\rightarrow$ `Linear(d_hidden)` $\rightarrow$ `SiLU()` $\rightarrow$ `Linear(2*d_latent)`; splits into $(\mu_t, \sigma_t)$ with $\sigma_t = $ `Softplus`$(\cdot)$; time-weighted encoder aggregation enabled; no explicit time input
- Prior initial distribution: learnable diagonal Gaussian in $\mathbb{R}^{d\_latent}$
- Prior drift network (autonomous): `Input(d_latent)` $\rightarrow$ `Linear(30)` $\rightarrow$ `Softplus()` $\rightarrow$ `Linear(d_latent)`
- Diffusion networks (diagonal, per-latent): `Input(1)` $\rightarrow$ `Linear(30)` $\rightarrow$ `Softplus()` $\rightarrow$ `Linear(1)` $\rightarrow$ `Sigmoid()`
- Decoder (observation model): `Input(d_latent)` $\rightarrow$ $2\times$[`Linear(30)` $\rightarrow$ `Softplus()`] $\rightarrow$ `Linear(d_obs)`; Gaussian likelihood with fixed std ($\sigma_0 = 1.0$, floor $10^{-3}$)

- Parameters:
  - Prior SDE: 1,550
  - Prior Observation: 2,860
  - Posterior Encoder: 3,154,000
  - Posterior Affine: 2,022,021

- Training:
  - Iterations: 100,000
  - Optimiser: AdamW
  - Learning rate: 0.0001
  - KL weight: 1.0
  - Sequence lengths: train=200, test=300;
  - Evaluation: Euler–Maruyama integration with fixed step $\Delta t = 0.01$

### E.3.6 Evaluation and Metrics

We report the mean squared error (MSE) between the predicted mean trajectory and the ground truth joint angles over the forecast horizon (steps 4–300 for Setup 1, 101–300 for Setup 2). Results in Table 3 are averaged over 10 runs with different random seeds, reporting mean $\pm$ 95% t-confidence intervals where available from original papers or our runs.

### E.3.7 Detailed Results

Table 8 provides FLOPs and runtime comparisons. Despite comparable computational complexity (FLOPs), Latent NSF achieves significantly faster runtime due to its ability to predict states at arbitrary time points in parallel, unlike traditional SDE methods that require sequential time-stepping.

## E.4 Stochastic Moving MNIST

### E.4.1 Data Generation

Data generation for Stochastic Moving MNIST was as follows:

- Frame size: 64x64 pixels
- Sequence length: 25 steps
- Digits: Two MNIST digits, chosen randomly
- Initial conditions: Positions and velocities sampled uniformly at random
- Bouncing mechanism: Modified from Denton and Fergus [11] for perfect reflection off boundaries with small angular noise $\mathcal{N}(0, \sigma_{\text{bounce}}^2)$, where $\sigma_{\text{bounce}} = 0.1$ rad
- Dataset size:
  - Training: 60,000 sequences (using original MNIST training set)
  - Test: 10,000 sequences (using MNIST test set)

Table 7: Runtime comparison (ms) for Stochastic Lorenz Attractor experiments. All measurements are mean times for generating a batch of 100 vectorised samples (10 trials). EM refers to Euler–Maruyama method, and SRK refers to Stochastic Runge–Kutta method.

| Method | Framework | $t = 0.25$ | $t = 0.5$ | $t = 0.75$ | $t = 1.0$ |
|---|---|---|---|---|---|
| Latent SDE (EM, $\Delta t = 0.01$) | PyTorch | 141.7 | 281.8 | 417.8 | 1,988.3 |
| Latent SDE (EM, $\Delta t = 0.01$) | JAX | 123.5 | 130.7 | 150.6 | 147.8 |
| Latent SDE (Milstein, $\Delta t = 0.01$) | PyTorch | 493.4 | 951.2 | 1,449.9 | 3,294.2 |
| Latent SDE (Milstein, $\Delta t = 0.01$) | JAX | 142.7 | 149.4 | 181.4 | 188.7 |
| Latent SDE (SRK, $\Delta t = 0.01$) | PyTorch | 1,084.5 | 2,189.0 | 3,283.9 | 5,827.9 |
| Neural LSDE (EM, $\Delta t = 0.01$) | PyTorch | 101.0 | 198.9 | 291.6 | 1,632.7 |
| Neural LSDE (Milstein, $\Delta t = 0.01$) | PyTorch | 158.4 | 250.9 | 396.1 | 1,767.9 |
| Neural LSDE (SRK, $\Delta t = 0.01$) | PyTorch | 886.5 | 1,799.9 | 2,681.0 | 4,807.4 |
| Neural GSDE (EM, $\Delta t = 0.01$) | PyTorch | 107.6 | 215.4 | 316.9 | 1,662.3 |
| Neural GSDE (Milstein, $\Delta t = 0.01$) | PyTorch | 137.7 | 274.9 | 431.9 | 1,852.9 |
| Neural GSDE (SRK, $\Delta t = 0.01$) | PyTorch | 964.2 | 1,962.1 | 2,925.3 | 5,155.3 |
| Neural LNSDE (EM, $\Delta t = 0.01$) | PyTorch | 104.7 | 209.7 | 310.9 | 1,644.0 |
| Neural LNSDE (Milstein, $\Delta t = 0.01$) | PyTorch | 130.8 | 290.7 | 381.9 | 1,779.1 |
| Neural LNSDE (SRK, $\Delta t = 0.01$) | PyTorch | 950.0 | 1,921.4 | 2,853.7 | 5,023.7 |
| NSF ($H_{\text{pred}} = 0.25$) | JAX | 0.221 | 0.303 | 0.341 | 0.373 |
| NSF ($H_{\text{pred}} = 0.5$) | JAX | 0.229 | 0.287 | 0.332 | 0.310 |
| NSF ($H_{\text{pred}} = 1.0$) | JAX | 0.255 | 0.286 | 0.294 | 0.295 |

Table 8: FLOPs, runtime, and MSE comparison for CMU Motion Capture experiments. FLOPs are measured per sample (latent SDE uses Euler–Maruyama integration), while runtime measurements represent the time required to generate a batch of 100 vectorised samples.

| Method | Framework | FLOPs (M) | | Runtime (ms) | | MSE | |
|---|---|---|---|---|---|---|---|
| | | Setup 1 | Setup 2 | Setup 1 | Setup 2 | Setup 1 | Setup 2 |
| Latent SDE ($\Delta t = 0.005$) | JAX | 13.5 | 25.1 | 3,585.0 | 3,236.0 | – | – |
| Latent SDE ($\Delta t = 0.005$) | PyTorch | 13.5 | 25.1 | 10,254.9 | 17,039.7 | 13.0 | 10.2 |
| Latent SDE ($\Delta t = 0.01$) | JAX | 8.0 | 17.8 | 1,894.0 | 1,664.0 | – | – |
| Latent SDE ($\Delta t = 0.01$) | PyTorch | 8.0 | 17.8 | 5,338.7 | 8,594.0 | 12.9 | 10.1 |
| Latent SDE ($\Delta t = 0.02$) | JAX | 5.2 | 14.1 | 854.2 | 909.6 | – | – |
| Latent SDE ($\Delta t = 0.02$) | PyTorch | 5.2 | 14.1 | 2,663.2 | 4,455.8 | 13.0 | 10.1 |
| Latent SDE ($\Delta t = 0.05$) | JAX | 3.6 | 11.9 | 374.1 | 494.2 | – | – |
| Latent SDE ($\Delta t = 0.05$) | PyTorch | 3.6 | 11.9 | 1,201.2 | 2,028.2 | 12.8 | 10.2 |
| Latent SDE ($\Delta t = 0.1$) | JAX | 3.0 | 11.2 | 223.0 | 372.6 | – | – |
| Latent SDE ($\Delta t = 0.1$) | PyTorch | 3.0 | 11.2 | 595.1 | 1,155.9 | 12.9 | 9.7 |
| Latent SDE ($\Delta t = 0.2$) | JAX | 2.7 | 10.8 | 148.4 | 323.3 | – | – |
| Latent SDE ($\Delta t = 0.2$) | PyTorch | 2.7 | 10.8 | 279.6 | 764.6 | 12.9 | 10.6 |
| Latent SDE ($\Delta t = 0.5$) | JAX | 2.6 | 10.6 | 107.1 | 289.6 | – | – |
| Latent SDE ($\Delta t = 0.5$) | PyTorch | 2.6 | 10.6 | 136.3 | 584.4 | 12.8 | 31.3 |
| Latent SDE ($\Delta t = 1$) | JAX | 2.5 | 10.5 | 99.88 | 279.6 | – | – |
| Latent SDE ($\Delta t = 1$) | PyTorch | 2.5 | 10.5 | 95.4 | 522.9 | 13.3 | 281.3 |
| Latent SDE ($\Delta t = 2$) | JAX | 2.5 | 10.5 | 92.19 | 266.1 | – | – |
| Latent SDE ($\Delta t = 2$) | PyTorch | 2.5 | 10.5 | 77.9 | 492.7 | 19.9 | – |
| Latent SDE ($\Delta t = 5$) | JAX | 2.5 | 10.5 | 89.13 | 269.7 | – | – |
| Latent SDE ($\Delta t = 5$) | PyTorch | 2.5 | 10.5 | 64.4 | 474.6 | 362.1 | – |
| Latent NSF (ours) | JAX | 6.0 | 17.7 | 0.44 | 1.8 | 8.7 | 3.4 |

### E.4.2 Configurations for Latent NSF

- Architecture:
  - Latent dimension `d_latent = 10`; per-frame embedding dimension `d_embed = 64`; scene dimension `d_scene = 64`
  - Per-frame encoder (CNN, $64 \times 64 \rightarrow 4 \times 4$): 4 down blocks: `Conv2d(3*3, stride=1, padding=1)` $\rightarrow$ `MaxPool2d(2,2)` $\rightarrow$ `GroupNorm(8)` $\rightarrow$ `SiLU()`; channels: $1 \rightarrow 64 \rightarrow 128 \rightarrow 256 \rightarrow 256$; spatial: $64 \rightarrow 32 \rightarrow 16 \rightarrow 8 \rightarrow 4$; head: `Flatten()` $\rightarrow$ `Linear(d_embed)`
  - Posterior (inference network): `Input(d_embed)` $\rightarrow$ `GRU(128)`; head: `Input(128)` $\rightarrow$ `Linear(128)` $\rightarrow$ `SiLU()` $\rightarrow$ `Linear(2*d_latent)`; splits into (`mean`, `std`) with `std` via `Softplus()`
  - Scene encoder: temporal median pooling over per-frame embeddings; `Input(d_embed)` $\rightarrow$ `Linear(128)` $\rightarrow$ `SiLU()` $\rightarrow$ `Linear(d_scene)`
  - Emission: inputs [scene, x_t] with size `d_scene + d_latent`; `MLP()` $\rightarrow$ `Linear(4*4*4*d_embed)`; reshape to (`4*d_embed, 4, 4`); 4 up blocks: `Conv(3*3)` $\rightarrow$ `GroupNorm(8)` $\rightarrow$ `upsample(*2)` $\rightarrow$ `SiLU()`, channels: `4*d_embed` $\rightarrow$ `4*d_embed` $\rightarrow$ `2*d_embed` $\rightarrow$ `d_embed`; spatial: $4 \rightarrow 8 \rightarrow 16 \rightarrow 32$; then `Conv(3*3)` $\rightarrow$ `SiLU()` $\rightarrow$ `Conv(3*3)` $\rightarrow$ `Sigmoid()`; output size `C=1, H=W=64`; Gaussian likelihood with uniform trainable std (shared across pixels)
  - NSF transition model (latent prior):
    * Gaussian base: `Input()` $\rightarrow$ $2 \times$`[Linear(64)` $\rightarrow$ `SiLU()]` $\rightarrow$ `Linear(2*d_latent)`; splits into (`mean`, `std`) with `std` via `Softplus()`
    * Affine coupling flow: 4 layers with alternating masking (autonomous SDE)
    * Per-layer conditioner: `Input(d_latent/2)` $\rightarrow$ $2 \times$`[Linear(64)` $\rightarrow$ `SiLU()]` $\rightarrow$ `Linear(d_latent)`; splits into (`scale`, `shift`)
  - Bridge model:
    * Gaussian base: `Input()` $\rightarrow$ $2 \times$`[Linear(64)` $\rightarrow$ `SiLU()]` $\rightarrow$ `Linear(2*d_latent)`; splits into (`mean`, `std`) with `std` via `Softplus()`
    * Affine coupling flow: 4 layers with alternating masking
    * Per-layer conditioner: `Input(d_latent)` $\rightarrow$ $2 \times$`[Linear(64)` $\rightarrow$ `SiLU()]` $\rightarrow$ `Linear(d_latent)`

- Parameters:
  - Encoder: 1,223,360
  - Decoder: 1,341,570
  - Posterior: 110,228
  - Scene encoder: 33,088
  - NSF prior: 33,740
  - Bridge model: 37,260

- Training:
  - Epochs: 1,000
  - Batch size: 64
  - Learning rate: 0.001
  - Latent NSF objective parameters: $\beta = 30.0$, $\lambda = 0.1$, $\beta_{\text{skip}} = 30.0$, $H_{\text{train}} = 2.5$
  - Warm-up: $\beta$, $\beta_{\text{skip}}$, and flow loss weight linearly increased over the first 20 epochs
  - Auxiliary updates: $K = 3$ inner steps per iteration for the bridge model
  - Skip-ahead KL: up to 25-step horizons; 25 samples per skip

### E.4.3 Configurations for Baseline

**Latent SDE** We used an implementation based on Daems et al. [10]. The baseline latent SDE model characteristics are as follows:

- Architecture:

- Latent dimension `d_latent = 10`; per-frame embedding dimension `d_embed = 64`; content dimension `d_contents = 64`
- Prior drift network: `Input(d_latent)` → `Linear(200)` → `Tanh()` → `Linear(200)` → `Tanh()` → `Linear(d_latent)`
- Diffusion networks (per-latent): `Input(1)` → `Linear(200)` → `Tanh()` → `Linear(200)` → `Tanh()` → `Linear(1)` → `Softplus()`
- Posterior control network: inputs `[x, y, h(t)]`; `Input(2*d_latent + d_embed)` → `Linear(200)` → `Tanh()` → `Linear(200)` → `Tanh()` → `Linear(d_latent)`; last layer kernel initialised to zero
- Encoder (2D conv, frames 64*64*1): four down blocks: `Conv(3*3)` → `MaxPool(2)` → `GroupNorm(8)` → `SiLU()`; channels 1 → `d_embed` → `2*d_embed` → `4*d_embed` → `4*d_embed`; spatial 64 → 32 → 16 → 8 → 4; head: `Flatten()` → `Linear(d_embed)` per frame
- Content extractor: temporal median pooling over per-frame embeddings; `Input(d_embed)` → `Linear(d_embed)` → `SiLU()` → `Linear(d_contents)`; outputs `w`
- Inference network for `x_0`: temporal `Conv(3)` → `SiLU()` → `Conv(3)`; then `MLP()` on concatenated features to output `2*d_latent` (mean and log-variance)
- Decoder: inputs `[w, x_t]` with size `d_contents + d_latent`; `MLP()` → `Linear(4*4*4*d_embed)`; reshape to (4, 4, 4*d_embed); 4 up blocks: `Conv(3*3)` → `GroupNorm(8)` → upsample(*2) → `SiLU()`, with channels `4*d_embed` → `4*d_embed` → `2*d_embed` → `d_embed`; spatial 4 → 8 → 16 → 32; then `Conv(3*3)` → `SiLU()` → `Conv(3*3)` → `Sigmoid()`; output size C=1, H=W=64

- Parameters:
  - Encoder: 1,269,972
  - Scene encoder: 8,320
  - Decoder: 1,341,569
  - Prior drift network: 44,410
  - Posterior drift network: 59,210
  - Diffusion network: 408,010
- Training:
  - Epochs: 100
  - Batch size: 32
  - Optimiser: Adam
  - Learning rate: 0.0003
  - KL weight: 0.1
  - Numerical integration: Stratonovich–Milstein with step size equal to one third of the observation interval

### E.4.4 Evaluation and Metrics

Evaluation was performed using Fréchet distances with embeddings from the pre-trained SRVP model [14] (using publicly available weights). The computed metrics are:

- Static FD: Based on time-averaged frame embeddings
- Dynamics FD: Based on sequence-level dynamics embeddings
- Frame-wise FD: Averaged per-frame embedding distances

Validation of these metrics is provided in Appendix G.

### E.4.5 Detailed Results

Table 9 provides computational comparisons for the Stochastic Moving MNIST experiments. Beyond the quantitative metrics reported in the main text, we provide qualitative visualisations to demonstrate

the reconstruction and prediction quality of our Latent NSF model compared to the latent SDE baseline.

Figure 6 shows a direct comparison of reconstruction quality between latent SDE and Latent NSF on input sequences. Both models are tasked with encoding and then reconstructing the same 25-frame input sequences. The figure demonstrates that Latent NSF achieves comparable reconstruction quality while maintaining computational efficiency.

For predictive performance, Figure 7 illustrates the models' ability to forecast future frames given the first 25 frames of a sequence. The ground truth trajectory (top row) shows the natural bouncing dynamics of the two MNIST digits. The latent SDE predictions (middle row) and Latent NSF recursive predictions (bottom row) both capture the general trajectory and bouncing behaviour, though with different levels of visual fidelity and temporal consistency.

Finally, Figure 8 provides a focused comparison at a specific prediction horizon (20 steps into the future) by showing multiple predicted samples from each model. This visualisation allows for assessment of both the accuracy of individual predictions and the diversity of the stochastic predictions across different model variants.

## F  Hyperparameter Sensitivity and Ablation Studies

To validate the sensitivity of the hyperparameters and effectiveness of the flow loss, we conduct an ablation study on the Stochastic Lorenz Attractor with missing data (Section F.1), and CMU Motion Capture dataset with extrapolation task (Setup 2) (Section F.2).

### F.1  Stochastic Lorenz Attractor with Missing Data

We trained on Stochastic Lorenz Attractor data where only a random continuous segment of length 0.5 from the full trajectory $t \in [0, 1]$ was observed.

#### F.1.1  Experimental Setup

We modify the training protocol as follows:

- **Original data structure**: The complete dataset contains trajectories with 41 time points at $t \in \{0, 0.025, 0.05, \ldots, 0.975, 1.0\}$, sampled at regular intervals of $\Delta t = 0.025$.

- **Missing data protocol**: We only use random continuous segment of length 20 time steps from the training data.

- **Training pairs**: From the 20 time points, we construct all possible state transition pairs $(x_{t_i}, x_{t_j})$ where $t_i < t_j$. This yields $20 \times 19/2 = 190$ unique pairs per trajectory, compared to 820 pairs in the complete data setting. Crucially, no training pairs directly connect states across $> 20$ time steps.

- **Training conditions**: All other hyperparameters remain identical to Section E.2.2:
  - Epochs: 1000
  - Optimiser: AdamW with learning rate 0.001 and weight decay $10^{-5}$
  - Batch size: 256
  - Inner steps for auxiliary model: $K = 5$ inner optimisation steps
  - Flow loss weights: $\lambda = 0.4$
  - Flow loss balancing: $\lambda_{\text{1-to-2}} = \lambda_{\text{2-to-1}} = 1.0$

- **Evaluation**: We compute KL divergence between the model and true distributions at $t \in \{0.5, 1.0\}$ using kernel density estimation as described in Section E.2.4. Results are averaged over 5 random seeds.

Based on this baseline condition, we conducted experiments varying the inner steps $K$, flow loss weight $\lambda$, and directional flow loss components $\lambda_{\text{1-to-2}}$ and $\lambda_{\text{2-to-1}}$.

Table 9: FLOPs and runtime comparison for Stochastic Moving MNIST experiments, showing computational costs for encoding 25 input frames and predicting the subsequent 25 frames. Runtime measurements represent the time required to process a batch of 100 vectorised samples. SM: Stratonovich–Milstein solver.

| Method | Framework | FLOPs (M) | Runtime (ms) |
|---|---|---|---|
| Latent SDE (SM) | JAX | 20,264 | 750 |
| Latent NSF (recursive) | JAX | 20,276 | 338 |
| Latent NSF (one-step) | JAX | 20,277 | 358 |

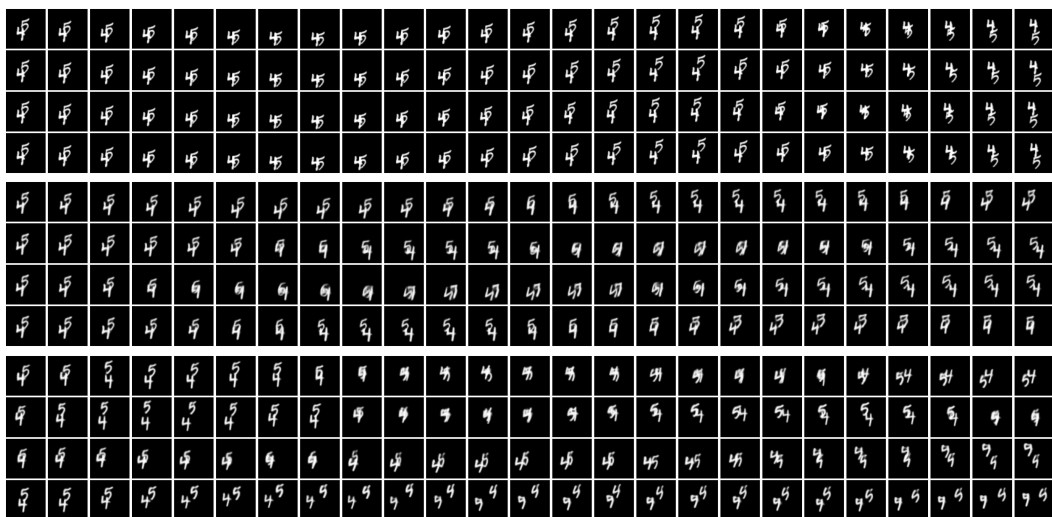

Figure 6: Reconstruction quality comparison on Stochastic Moving MNIST. Top row: original input frames, middle row: latent SDE reconstructions, bottom row: Latent NSF (recursive) reconstructions. Each column shows consecutive time steps (frames 1–25). Both models successfully capture the appearance and positioning of the bouncing digits.

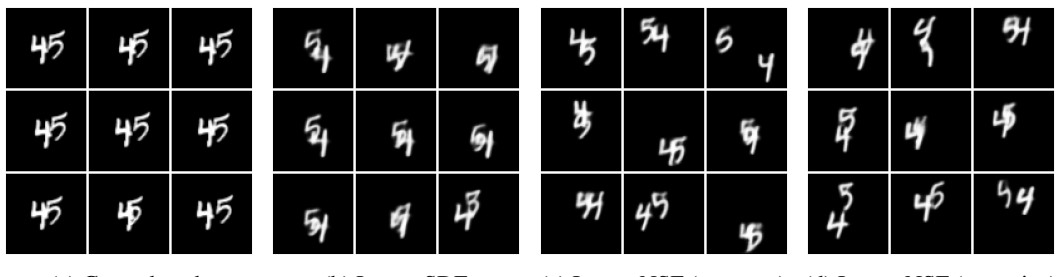

Figure 7: Future frame prediction comparison on Stochastic Moving MNIST. Top: ground truth continuation, middle: latent SDE predictions, bottom: Latent NSF predictions. Both models predict 25 future frames given the first 25 frames. The comparison shows how well each model captures stochastic bouncing dynamics while maintaining visual quality and temporal coherence.

(a) Ground truth      (b) Latent SDE      (c) Latent NSF (one-step)    (d) Latent NSF (recursive)

Figure 8: Stochastic prediction samples 20 steps into the future on Stochastic Moving MNIST. Each panel shows multiple independent samples generated after observing the first 25 frames and predicting 20 steps ahead.

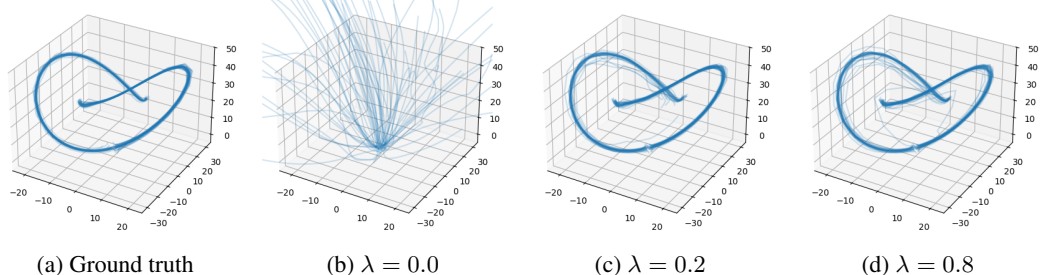

|  (a) Ground truth | (b) $\lambda = 0.0$ | (c) $\lambda = 0.2$ | (d) $\lambda = 0.8$ |

Figure 9: Comparison of generated samples (64 samples per panel) on the stochastic Lorenz attractor. Columns show ground truth and model samples with different flow loss weights $\lambda \in \{0.0, 0.2, 0.8\}$.

Table 10: Flow loss ablation on missing Stochastic Lorenz. Reported values are mean $\pm$ standard deviation over seeds.

| Flow loss weight $(\lambda)$ | KL $(t = 0.5)$ | KL $(t = 1.0)$ |
|---|---|---|
| 0.0 (no flow loss) | $7.9 \pm 0.2$ | $22.9 \pm 1.8$ |
| 0.001 | $3.0 \pm 0.2$ | $4.5 \pm 0.9$ |
| 0.01 | $1.7 \pm 0.1$ | $2.1 \pm 0.6$ |
| 0.4 (default) | $1.2 \pm 0.1$ | $1.2 \pm 1.0$ |
| 0.8 | $1.3 \pm 0.2$ | $1.3 \pm 0.6$ |

Table 11: Effect of auxiliary optimisation steps $K$ on KL and training time (per epoch). $\lambda = 0.4$. Simultaneous: auxiliary and main models updated jointly without inner loops, as described in Appendix C.1.

| K (auxiliary inner steps) | KL $(t = 0.5)$ | KL $(t = 1.0)$ | Epoch time (s) |
|---|---|---|---|
| 0 (no auxiliary steps) | $7.9 \pm 0.2$ | $22.9 \pm 1.8$ | 66 |
| 0 (simultaneous) | $1.1 \pm 0.1$ | $1.8 \pm 1.2$ | 106 |
| 1 | $1.1 \pm 0.1$ | $1.7 \pm 0.6$ | 125 |
| 5 (default) | $1.2 \pm 0.1$ | $1.4 \pm 0.6$ | 201 |
| 10 | $1.1 \pm 0.2$ | $1.2 \pm 0.7$ | 295 |

Table 12: Effect of directional flow loss components on KL. $\lambda = 0.4, K = 5$.

| Flow loss direction | KL $(t = 0.5)$ | KL $(t = 1.0)$ |
|---|---|---|
| 1-to-2 only $\lambda_{\text{1-to-2}} = 1, \lambda_{\text{2-to-1}} = 0$ | $1.1 \pm 0.1$ | $1.4 \pm 0.6$ |
| 2-to-1 only $\lambda_{\text{1-to-2}} = 0, \lambda_{\text{2-to-1}} = 1$ | $1.2 \pm 0.1$ | $1.4 \pm 0.7$ |
| Bidirectional $\lambda_{\text{1-to-2}} = \lambda_{\text{2-to-1}} = 1$ | $1.2 \pm 0.1$ | $1.4 \pm 0.6$ |

### F.1.2 Results

Flow loss weight sensitivity is summarised in Table 10, and visualisation of 64 trajectories for the ground truth and generated samples is shown in Figure 9, which is obtained in the same way as Figure 3 in the main text with one-step prediction. The effect of auxiliary inner steps is shown in Table 11. Directional components are compared in Table 12. Overall, the sensitivity to the flow loss weight and the number of inner steps is low when the order of magnitude is around the baseline conditions.

## F.2 CMU Motion Capture Dataset with Extrapolation Task (Setup 2)

### F.2.1 Experimental Setup

We use the same experimental setup as Setup 2 in Section E.3.4, and conducted the parametric study varying the inner steps $K$, flow loss weight $\lambda$, and directional flow loss components $\lambda_{\text{1-to-2}}$ and $\lambda_{\text{2-to-1}}$ on the CMU Motion Capture dataset with extrapolation task (Setup 2).

### F.2.2 Results

Flow loss weight (bidirectional, $K = 3$) is reported in Table 13. Directional variants for flow loss at $\lambda = 1.0$, $K = 3$ are summarised in Table 14. The number of inner steps for the auxiliary model (with $\lambda = 1.0$, bidirectional) are shown in Table 15. In short, best MSE occurs near $\lambda = 0.03$, direction 2-to-1 is slightly better than 1-to-2 but bidirectional is comparable, and small $K$ (e.g., 1) also works while $K = 0$ fails, which means that, in this case, the sensitivity to the flow loss weight and the number of inner steps is low when the order of magnitude is around the baseline conditions.

## F.3 Practical Guidance on Hyperparameters

We provide practical guidance on hyperparameters specifically for the NSF or Latent NSF.

### F.3.1 Time Horizon for Training $H_{\text{train}}$ and Inference $H_{\text{pred}}$

The time horizon for training should be set to the maximum one-shot interval used during training. In our experiments, we found that there is no significant difference in performance dependent on the $H_{\text{pred}}$ value. However we have not explored the effect of the $H_{\text{train}}$ value. Since, in practice, large $H_{\text{train}}$ requires more expressive model capacity, we recommend balancing trade-off between model capacity and prediction runtime using $H_{\text{train}}$, and using $H_{\text{pred}} = H_{\text{train}}$ in most cases, depending on the data complexity and prediction runtime requirements.

### F.3.2 Flow Loss Weight

For any experiment we have conducted, we found that sub-order of magnitude $\lambda$ values are sufficient to achieve good performance, assuming the data is normalised.

### F.3.3 Auxiliary Inner Steps or Simultaneous Updates

The auxiliary inner steps $K$ controls the number of inner steps for the auxiliary model. We recommend using simultaneous updates first, as described in Appendix C.2, and if there is training instability, we recommend using $K > 0$ and increasing the value of $K$ until the flow loss is stable.

### F.3.4 Directional Flow Loss Components

The directional flow loss components $\lambda_{\text{1-to-2}}$ and $\lambda_{\text{2-to-1}}$ control the strength of the flow loss in the 1-to-2 and 2-to-1 directions respectively. In our experiments, though the performance is not significantly different, we recommend using bidirectional flow loss for symmetry and stability.

# G  Validation of the Fréchet Image and Video Metrics

## G.1  Protocol

To confirm that the three Fréchet distances introduced in §6.3 (static content, dynamics and frame-wise) behave as intended on our *Stochastic Moving MNIST* benchmark, we performed a series of controlled experiments. Code for reproducing the experiments is available at `https://github.com/nkiyohara/srvp-fd`.

### G.1.1  Datasets.

For each trial, two datasets were instantiated as follows:

- Content: Each containing 256 sequences of 25 frames.

- Production method: Same as in the main text with identical hyperparameters, varying only specific factors (digit classes, handwriting samples, or dynamics).

- Digit ordering: To remove ambiguity, a second copy of the first dataset was generated with digit indices swapped. The version yielding the smaller static Fréchet distance was retained.[1]

### G.1.2  Conditions.

Four experimental conditions were repeated over 32 trials (each with a fresh random seed):

- **Different dynamics**: same digits and handwriting, different dynamics.

- **Different digits**: different digit classes, identical dynamics.

- **Different handwriting**: same digit classes, different handwriting, identical dynamics.

- **Different both**: different digit classes *and* different dynamics.

### G.1.3  Computation of distances.

Distance computation details:

- Encoder: Public SRVP encoder of Franceschi et al. [14].

- Frame-wise scores: Averaged the 25 frame distances to yield a single number per sequence pair.

## G.2  Results

Table 16 reports the mean and standard deviation (over the 32 trials) of each metric. The metrics reflect the intended factors:

- Varying only the dynamics (*different dynamics*) drives the **dynamics** distance up by an order of magnitude while leaving the static distance low.

- Altering appearance while keeping motion fixed (*different digits/handwriting*) raises the **static** distance, with digits > handwriting as expected, and leaves the dynamics distance low.

- When both factors change (*different both*), *both* metrics are simultaneously high.

- Frame-wise distances stay near zero unless both content and motion differ, indicating that they act as a weak sanity check.

---

[1]Without this post-hoc swap the static metric can be artificially inflated when the same digits appear in reverse order.

Table 13: Effect of flow loss weight $\lambda$ on test MSE (mean $\pm$ 95% confidence interval in t-statistic) for CMU Motion Capture (Setup 2). Auxiliary inner steps are fixed to $K = 3$. Flow loss weight $\lambda = 1.0$, bidirectional($\lambda_{\text{1-to-2}} = \lambda_{\text{2-to-1}} = 1.0$), and KL weight $\beta = \beta_{\text{skip}} = 0.3$.

| Flow loss weight ($\lambda$) | MSE |
|---|---|
| 0.001 | $4.08 \pm 0.46$ |
| 0.003 | $3.77 \pm 0.37$ |
| 0.01 | $3.73 \pm 0.48$ |
| 0.03 | $3.36 \pm 0.25$ |
| 0.1 | $3.43 \pm 0.29$ |
| 0.3 | $3.60 \pm 0.33$ |
| 1.0 | $3.41 \pm 0.27$ |

Table 14: Effect of directional flow loss components on test MSE (mean $\pm$ 95% confidence interval in t-statistic) for CMU Motion Capture (Setup 2). Flow loss weight $\lambda = 1.0$, inner loop steps for bridge model $K = 3$, and KL weight $\beta = \beta_{\text{skip}} = 0.3$.

| Flow loss direction | MSE |
|---|---|
| 1-to-2 only $\lambda_{\text{1-to-2}} = 1, \lambda_{\text{2-to-1}} = 0$ | $3.66 \pm 0.50$ |
| 2-to-1 only $\lambda_{\text{1-to-2}} = 0, \lambda_{\text{2-to-1}} = 1$ | $3.48 \pm 0.46$ |
| Bidirectional $\lambda_{\text{1-to-2}} = \lambda_{\text{2-to-1}} = 1$ | $3.41 \pm 0.27$ |

Table 15: Effect of auxiliary optimisation steps $K$ on test MSE (mean $\pm$ 95% confidence interval in t-statistic) for CMU Motion Capture (Setup 2). Flow loss weight $\lambda = 1.0$, bidirectional($\lambda_{\text{1-to-2}} = \lambda_{\text{2-to-1}} = 1.0$), and KL weight $\beta = \beta_{\text{skip}} = 0.3$. "Simultaneous" refers to updating auxiliary and main models jointly without inner loops for bridge model, as described in Appendix C.2.

| $K$ (auxiliary inner steps) | MSE |
|---|---|
| 0 (no auxiliary training) | $261.82 \pm 124.57$ |
| 0 (simultaneous) | $3.59 \pm 0.61$ |
| 1 | $3.35 \pm 0.39$ |
| 3 (default) | $3.41 \pm 0.27$ |
| 5 | $3.68 \pm 0.40$ |
| 10 | $3.59 \pm 0.38$ |

Table 16: Mean $\pm$ standard deviation of the three Fréchet distances across 32 trials.

| Condition | Static | Dynamics | Frame mean |
|---|---|---|---|
| Different dynamics | $2.57 \pm 1.40$ | $14.85 \pm 5.48$ | $0.80 \pm 0.14$ |
| Different digits | $5.66 \pm 1.82$ | $1.75 \pm 1.83$ | $0.13 \pm 0.06$ |
| Different handwriting | $3.74 \pm 1.65$ | $1.79 \pm 1.66$ | $0.09 \pm 0.05$ |
| Different both | $6.78 \pm 1.95$ | $14.74 \pm 5.79$ | $0.87 \pm 0.13$ |

### G.3 Discussion

The clear separation between conditions, the low variance within each condition and the alignment with human intuition together demonstrate that our Fréchet metrics are *both reliable and interpretable*. They therefore provide a sound basis for quantitative evaluation of generative video models on Stochastic Moving MNIST, capturing complementary aspects of visual content and motion without leakage between the two.

