# OpenReview forum: "Neural Stochastic Flows: Solver-Free Modelling and Inference for SDE Solutions"
_NeurIPS.cc/2025/Conference — NeurIPS 2025 poster_

### Official Review · Reviewer_yfA9 · 2025-06-18

**Clarity:** 2
**Significance:** 3
**Originality:** 3
**Rating:** 4
**Confidence:** 4

**Summary:**

The paper presents neural stochastic flows, a novel framework for modelling trajectories of stochastic differential equations for time series data. The method utilizes normalising flows to get a direct access to the probability distribution of SDE transitions, which allows it to obtain a sample from an arbitrary time step without requiring a numerical solver. Experimental results show that the proposed approach significantly speeds up the sampling process while maintaining high sample quality.

**Questions:**

All my questions are related to the Experiments section.

**Stochastic Lorenz Attractor**
1. In Table 1, the meaning of the variable $t$ is unclear. What does it represent, and why does the KL divergence vary non-monotonically with changes in $t$?
2. Why does NSF with $t_{\text{max}} = 0.5$ perform worse than NSF with $t_{\text{max}} = 1$ and $t_{\text{max}} = 0.25$?
3. How does Figure 3 relate to Table 1? Were the depicted trajectories generated in a one-step fashion, or sequentially by conditioning each point on the previous one?

**CMU Motion Capture**
1. In the Latent SDE paper [1] I see that the reported MSE for Setup 1 is 4.03, which is notably better than the result of your reproduction. Please explain this discrepancy.
2. Figure 4 (a) clearly indicates that MSE grows with increasing runtime for Latent NSF. How can you explain this behavior?
3. You attribute the better performance of Latent NSF in Setup 2 to the SDE's inductive bias. Could you please clarify the idea of inductive bias here and elaborate on why Setup 2 constitutes extrapolation while Setup 1 does not for this particular dataset?

**Stochastic Moving MNIST**
1. The variance of the Fréchet Distance in Table 3 and Figure 5 appears high relative to the mean differences between methods. This raises concerns about the statistical significance of the results. How many samples were used to compute these metrics, and what is the level of significance?
2. There is no analysis of the results. How can you explain that recursive NSF is much better then SDE in terms of Static FD and much worse in terms of Dynamics FD? How does the speed of recursive NSF compare with one-step NSF? Are the results aligned with expectations, and how do they compare with findings on the other datasets? Why does recursive NSF perform better for Moving MNIST, while appearing to underperform on the Motion Capture dataset?

**General questions**
1. I noticed that you use different dimensionality of latent space for different methods (Appendix E.3 and E.4), which might have an impact on quality. Could you please explain how you chose these values and all other hyperparameters, and how different methods compare in terms of the number of parameters?
2. All experiments are conducted on relatively small-scale problems (latent space dimensionality ≤ 10). Is the proposed method applicable to larger-scale settings, such as in the video domain?

[1] X. Li, T. Wong, R. T. Q. Chen, and D. Duvenaud. Scalable gradients for stochastic differential
equations. In Proceedings of the 23rd International Conference on Artificial Intelligence and
Statistics (AISTATS), 2020.

**Ethical Concerns:**

["NO or VERY MINOR ethics concerns only"]

**Final Justification:**

My main concerns were addressed by the authors in their rebuttal:

* The difference between Latent SDE results in the original paper and the authors' reproduction is explained by the well-known issue with the original Latent SDE implementation.
* Experiment setup became clear for the Stochastic Lorenz Attractor.
* The difference between Static FD and Dynamics FD is explained for the Stochastic Moving MNIST and the statistical significance of the results is proven.

**Limitations:**

The limitations are discussed.

**Paper Formatting Concerns:**

No major formatting issues.

**Quality:**

2

**Strengths And Weaknesses:**

**Strengths**
The paper structure is good, the proposed method is well-motivated and its theoretical justification is strong.

**Weaknesses**
The experimental evaluation lacks clarity and depth in analysis (see Questions section). Some of the important implementation details are not presented in the main text of the paper. For example, the architecture of the proposed normalizing flow is a crucial part of the approach and it should not be moved to the appendix.

---

> ### Author Rebuttal · Authors · 2025-07-30
>
> We thank all reviewers for constructive feedback for which we will include in revision. We are encouraged that reviewers recognise our work as addressing an important problem with a novel approach. Below we will first state our core contributions and address common concerns, and then answer reviewer specific questions.
> # Summary of contributions
> We propose Neural Stochastic Flows (NSF) as a solver-free method to directly learns the weak solution of an SDE from data. Key components include
>
> - Introduction of stochastic flows to the field of sequential deep generative models, enabling one-step computation of conditional distributions of an SDE at arbitrary time points,
> - A conditional normalisation flow that preserves important properties of stochastic flows,
> - The “flow loss” as a regulariser to enforce flow consistency.
>
> Reviewers TqQA and YJDb endorsed the novelty of our contributions on one-step SDE conditional distribution evaluations, noting NSF’s mathematical soundness and substantial speedups. When compared with baselines, Reviewer aZhw commended on NSF being significantly faster while achieving comparable performance regarding accuracies.
> # Common questions
> ## Scalability (YJDb, yfA9)
> Regarding computation, our architecture is based on affine coupling-based flows, which is scalable, e.g., our stochastic moving MNIST experiment is conducted on videos with frame size 64 x 64, and RealNVP [12] and Glow [27] (based on affine coupling flows) has been applied to images. Future work will incorporate our proposed constraints into more expressive normalising flow architectures, e.g., TarFlow [51].
> ## Clarifying inductive bias of SDEs in MoCap dataset (YJDb, yfA9)
> Nonlinear SDEs naturally capture real-world stochastic dynamics in human motion, where random perturbations interact nonlinearly with the system state. This generalises better than ODEs (deterministic) and linear SDEs (state‑independent noise).
>
> Among nonlinear SDE approaches, Latent SDE models rely on controlled SDEs to represent the posterior, which involves learning complex drift function, that is difficult to optimise. SDE Matching addresses challenges by learning posteriors conditioned on the observations, $p(x_t|o_{t_1},o_{t_2},\ldots)$, but must learn complex, time-varying distributions. In contrast, our Latent NSF takes a simpler yet effective way: directly modeling transition laws $p(x_{t_j}|x_{t_i})$, using normalising flows. This avoids needing both control-based SDEs and complex time-dependent posteriors.
> ## Hyperparameter Sensitivity and Ablation Studies (YJDb, aZhw)
> We conducted ablation studies on challenging partial observation tasks. We trained on Stochastic Lorenz Attractor data where only a random continuous segment of length 0.5 from the full trajectory [0,1] was observed.
>
> **Flow Loss Ablation on Missing Stochastic Lorenz**
> |Flow Loss Weight (λ)|KL (t=0.5)|KL (t=1.0)|
> |-|-|-|
> |0.0 (no flow loss)|7.9 ± 0.2|22.9 ± 1.8|
> |0.001|3.0 ± 0.2|4.5 ± 0.9|
> |0.01|1.7 ± 0.1|2.1 ± 0.6|
> |0.4 (default)|1.2 ± 0.1|1.2 ± 1.0|
> |0.8|1.3 ± 0.2|1.3 ± 0.6|
>
> **Effect of Auxiliary Optimisation Steps (λ=0.4)**
> |K (auxiliary steps)|KL (t=0.5)|KL (t=1.0)|Epoch Time (s)|
> |-|-|-|-|
> |0 (no auxiliary)|7.9 ± 0.2|22.9 ± 1.8|66|
> |0 (simultaneous)*|1.1 ± 0.1|1.8 ± 1.2|106|
> |1 |1.1 ± 0.1|1.7 ± 0.6|125|
> |5 (default)|1.2 ± 0.1|1.4 ± 0.6|201|
> |10|1.1 ± 0.2|1.2 ± 0.7|295|
>
> \* Simultaneous: auxiliary and main models updated jointly without inner loops
>
> **Directional KL Components (λ=0.4, K=5)**
>
> |Flow Loss Direction|KL (t=0.5)|KL (t=1.0)|
> |-|-|-|
> |Forward only (1→2)|1.1 ± 0.1|1.4 ± 0.6|
> |Reverse only (2→1)|1.2 ± 0.1|1.4 ± 0.7|
> |Bidirectional|1.2 ± 0.1|1.4 ± 0.6|
>
> Without flow loss (λ = 0), the model fails to generalise to unobserved regions, with KL ≈ 23 at $t=1.0$. Adding flow loss dramatically improves performance; with λ = 0.2–0.8, KL stays around 1.2 across time steps.
>
> Applying flow loss in only one direction already reduces KL divergence. However, since forward KL encourages broad coverage of the distribution, and reverse KL focuses on matching high-density regions, we decided to use both equally to balance these effects with symmetry.
>
> A similar trend was observed on MoCap setup 2 and we will include these ablations in the revision.
> ## Main paper’s content organisation (TqQA, yfA9)
> We will move key architectural details (NSF design, training algorithm) to the main text for clarity.
>
> ---
> # Specific questions from Reviewer yfA9
> Thank you for your valuable feedback. We address your concerns regarding experiments below.
> ## Stochastic Lorenz Attractor
> 1. The variable t represents the prediction time horizon from the initial state. Specifically, we evaluate the KL divergence between the true distribution $p(x_t|x_0)$ and our model's predicted distribution at times t ∈ {0.25, 0.5, 0.75, 1.0}. The non-monotonic KL divergence is natural for chaotic systems, since dynamics of chaotic attractors exhibit complex convergence/divergence patterns.
> 2. In this setting, NSF is trained to predict arbitrary time intervals within [0,1] in a one step. During inference, we can either make one-step predictions with $t_\mathrm{max}=1.0$, or use recursive predictions with smaller $t_\mathrm{max}$ values (e.g., 4 steps of 0.25 each to reach $t=1.0$).
> The optimal t_max is not trivial to determine a priori, as stated in our Limitations section. The model may have learned better representations at certain time scales during training, leading to this non-monotonic behaviour.
> 3. Figure 3(d) shows trajectories generated using one-step predictions from NSF. Each line in the visualisation shows an sample from the learned conditional distribution $p(x_t|x_0)$, where we predict from the initial state $x_0$ to time t in a one step, with the same random sample from Gaussian. Table 1 evaluates the same model but provides KL divergence. The $t_\mathrm{max}$ parameter indicates the maximum one-step prediction interval for NSF, e.g., $t_\mathrm{max}=0.5$ means that reductions up to $t=0.5$ are one-step; beyond $t=0.5$ requires two steps.
> ## MoCap
> 1. You are correct that Li et al. [32] reported MSE 4.03 for Setup 1, while ours shows 12.91. This difference stems from known reproducibility issues with the original Latent SDE implementation on the MoCap dataset. This issue is also raised by multiple researchers in public discussions (e.g., official GitHub repo issue trackers from 2022). To ensure fair comparison, we re-implemented the method using consistent data processing and evaluation procedures. Our reported values for Latent SDE (12.91 ± 2.90) are consistent with the accuracy-runtime curve in the Appendix and with other reproduction reports mentioned in these public discussions. We will add a footnote in the camera-ready version explicitly noting this reproducibility issue.
> 2. The increasing MSE with runtime for Latent NSF reflects a characteristic of our training approach. The model is trained across the full interval $[0, t_\mathrm{max}]$, it learns transitions for all time gaps, but it remains an open question about whether one-step or recursive predictions performs the best at inference. In MoCap, one-step prediction yields the best balance of accuracy and efficiency.
> 3. For explanation for the inductive bias, see common rebuttal. The key difference between Setup 1 and Setup 2 is in the temporal split. Setup 1 trains and tests within the same time range (interpolation), and Setup 2 trains only on steps 1–200 and tests on unseen future steps 201–300 (extrapolation), requiring generalisation beyond the training horizon.
> ## Stochastic Moving MNIST
> 1. For computing these metrics, we used 256 generated samples for each of 32 different input conditions. We agree that comparing distributions in such high-dimensional spaces through finite samples is inherently challenging, which contributes to the high variance observed. This is a fundamental limitation when evaluating generative models for high-dimensional video data. However, as in Appendix F, our validation experiments show that these metrics do capture meaningful differences between static and dynamic aspects of the generated videos, despite the high variance, providing qualitative insights into the relative performance of the methods.
> 2. The trade-off between Static FD and Dynamics FD is controlled by the β parameter, which balances reconstruction error and the KL divergence term. We agree that analysing how this trade-off changes with β for both Latent SDE and Latent NSF would provide valuable insights. If accepted, we will include this analysis in the camera-ready version's appendix, showing how different β values affect the Static/Dynamics FD balance for both methods.
> The speed comparison is provided in Table 7 (Appendix). The difference between recursive and one-step NSF is tiny since the computational bottleneck is the decoder forward pass, not the NSF.
> The performance differences relate to which prediction horizons achieve highest accuracy for each model and dataset. As in our limitations, determining optimal prediction horizons a priori remains challenging. The flow loss is designed to maintain consistent accuracy across all prediction horizons, but dataset-specific characteristics still influence which sampling strategy performs best.
> ## General questions
> 1. We followed the same dimensionality as baseline papers standard (Stochastic Lorenz Attractor: [32], MoCap: [2]). For Stochastic Moving MNIST, we followed the dataset's original paper [10].
> We conducted systematic searches for key hyperparameters. For β (KL weight) and flow loss weight, as is standard for sequential VAE, we performed logarithmic grid search (…, 0.1, 0.3, 1.0, …) for both ours and baselines.
> 2. Noted in the common rebuttal “Scalability”.
>
> ---
> ## References
> For 1-50, please refer to the main text.
>
> [51] Zhai et al., "Normalizing Flows are Capable Generative Models", ICML 2025.
>
> ---
> We hope this addresses your concerns and would be happy to provide any additional clarifications.

---

> > ### Comment · Area_Chair_8vQr · 2025-08-05
> >
> > Dear reviewer yfA9,
> >
> > After considering the rebuttal to your review and the other reviews/rebuttals, how and why has this affected your position on this submission? Please reply with an official comment (not just the mandatory acknowledgement) reflecting your current view, any follow-up questions/comments etc.
> >
> > Note the Aug 6 AoE deadline, make sure to respond in time for the authors to be able to submit a response if necessary.

---

> ### Comment · Reviewer_yfA9 · 2025-08-05
>
> I would like to thank the authors for their thorough answers to my questions. It cleared up most of my doubts about the paper, so I am increasing my score to 4.
> Nevertheless, if the paper is accepted, I insist that the authors add more details to the discussion about the difficulty of choosing the $t_{max}$ parameter. In particular, which factors might affect the optimal choice. I would also appreciate the addition of the number of parameters of each model right after its description in the appendix in order to convince the reader that all models in comparison have similar sizes.

---

> > ### Author Response · Authors · 2025-08-05
> >
> > Thank you for your follow-up and for increasing your score to 4. We greatly appreciate your thoughtful review and suggestions, which are invaluable in refining our paper.
> >
> > We confirm that, if the paper is accepted, we will address both of your requests in the camera-ready version:
> >
> > 1. We will expand the discussion in the main text and/or appendix on the challenges of selecting the $t_{\mathrm{max}}$ parameter, including factors that influence the optimal choice (e.g., dynamics, training stability).
> > 2. We will include the number of parameters for each model after the descriptions in the appendix to ensure transparency and facilitate fair comparisons across methods.
> >
> > In addition, we will incorporate all clarifications, analyses, and ablation results provided in our rebuttal into the camera-ready version, so that these points are explicitly documented for readers.
> >
> > Thank you again for your valuable input.

---

### Official Review · Reviewer_aZhw · 2025-06-27

**Clarity:** 2
**Significance:** 2
**Originality:** 2
**Rating:** 4
**Confidence:** 3

**Summary:**

The authors propose to use conditional normalizing flows to approximate transition densities in SDE. This yields a solver-free method to sample between arbitrary timepoints in a single step. "Neural stochastic flows (NSF)" is a conditional coupling flow aiming to estimate transition densities $p(x_t|x_s, s, t-s)$ of SDEs (for all s,t). The authors enforce that some known properties of them are architecturally embedded, and develop a regularization term to enforce the "flow" property. It's shown in experiments on synthetic SDEs and real-world tracking and video data that NSF stays distributionally accurate while strongly reducing computational cost.

**Questions:**

See weaknesses i.e.:
- How much does the regularization term help (compared to no regularization) ?
- Why forward and backward KL for regularization? This also introduces the "weighting" hyperparameters. As reverse and forward KL are often on quite different "scales," this might be important to "tune". Did the authors investigate just using the forward term (or just the backward term)?

**Ethical Concerns:**

["NO or VERY MINOR ethics concerns only"]

**Final Justification:**

The authors adressed my main concerns in the rebutal:
- "Flow Loss Ablation" in the rebutal did clearly demonstrate that the auxiliary loss is important.
- The authors did to what is possible to include a fair comparission with SDE matching.

**Limitations:**

yes

**Quality:**

3

**Strengths And Weaknesses:**

Strengths:
- Well written, structured, and motivated, i.e., first pointing out the target's properties, then incorporating them in the model.
- Overall, the empirical results show that the proposed method is accurate and comparable to other approaches, but much more computationally efficient.

Weaknesses:
- One of the main methodological novelties is the regularization term for the Flow property. However, it only occupies ~0.5 page in the method section and some design decisions about it remain unclear (i.e., "A natural design for such a regularization loss is to combine both forward and reverse KL ..." Why is this the natural choice? There are several other possibilities to achieve the same.
- Missing ablations e.g. given that this regularization term adds quite a lot of complexity to the approach, I miss an ablation that this term does empirically improve upon the standard maximum likelihood loss (which should enforce the Flow Property implicitly in a data-driven manner). How much slower is it to train with regularization? I expect quite a bit.
- Table 2 raises questions: SDE Matching performs best in Setup 1 (by quite a big margin compared to the proposed method), but results for this approach on Setup 2 are not reported (why?). Given that the authors claim "state-of-the-art long-horizon extrapolation performance" this result should be included.

---

> ### Author Rebuttal · Authors · 2025-07-30
>
> We thank all reviewers for constructive feedback for which we will include in revision. We are encouraged that reviewers recognise our work as addressing an important problem with a novel approach. Below we will first state our core contributions and address common concerns, and then answer reviewer specific questions.
> # Summary of contributions
> We propose Neural Stochastic Flows (NSF) as a solver-free method to directly learns the weak solution of an SDE from data. Key components include
>
> - Introduction of stochastic flows to the field of sequential deep generative models, enabling one-step computation of conditional distributions of an SDE at arbitrary time points,
> - A conditional normalisation flow that preserves important properties of stochastic flows,
> - The “flow loss” as a regulariser to enforce flow consistency.
>
> Reviewers TqQA and YJDb endorsed the novelty of our contributions on one-step SDE conditional distribution evaluations, noting NSF’s mathematical soundness and substantial speedups. When compared with baselines, Reviewer aZhw commended on NSF being significantly faster while achieving comparable performance regarding accuracies.
> # Common questions
> ## Scalability (YJDb, yfA9)
> Regarding computation, our architecture is based on affine coupling-based flows, which is scalable, e.g., our stochastic moving MNIST experiment is conducted on videos with frame size 64 x 64, and RealNVP [12] and Glow [27] (based on affine coupling flows) has been applied to images. Future work will incorporate our proposed constraints into more expressive normalising flow architectures, e.g., TarFlow [51].
> ## Clarifying inductive bias of SDEs in MoCap dataset (YJDb, yfA9)
> Nonlinear SDEs naturally capture real-world stochastic dynamics in human motion, where random perturbations interact nonlinearly with the system state. This generalises better than ODEs (deterministic) and linear SDEs (state‑independent noise).
>
> Among nonlinear SDE approaches, Latent SDE models rely on controlled SDEs to represent the posterior, which involves learning complex drift function, that is difficult to optimise. SDE Matching addresses challenges by learning posteriors conditioned on the observations, $p(x_t \mid o_{t_1}, o_{t_2}, \ldots)$, but must learn complex, time-varying distributions. In contrast, our Latent NSF takes a simpler yet effective way: directly modeling transition laws $p(x_{t_j}\mid x_{t_i})$, using normalising flows. This avoids needing both control-based SDEs and complex time-dependent posteriors.
> ## Hyperparameter Sensitivity and Ablation Studies (YJDb, aZhw)
> We conducted ablation studies on challenging partial observation tasks. We trained on Stochastic Lorenz Attractor data where only a random continuous segment of length 0.5 from the full trajectory $[0,1]$ was observed.
>
> **Flow Loss Ablation on Missing Stochastic Lorenz**
> |Flow Loss Weight ($\lambda$)|KL ($t=0.5$)|KL ($t=1.0$)|
> |-|-|-|
> |0.0 (no flow loss)|7.9 ± 0.2|22.9 ± 1.8|
> |0.001|3.0 ± 0.2|4.5 ± 0.9|
> |0.01|1.7 ± 0.1|2.1 ± 0.6|
> |0.4 (default)|1.2 ± 0.1|1.2 ± 1.0|
> |0.8|1.3 ± 0.2|1.3 ± 0.6|
>
> **Effect of Auxiliary Optimisation Steps ($\lambda=0.4$)**
> |K (auxiliary steps)|KL ($t=0.5$)|KL ($t=1.0$)|Epoch Time (s)|
> |-|-|-|-|
> |0 (no auxiliary)|7.9 ± 0.2|22.9 ± 1.8|66|
> |0 (simultaneous)*|1.1 ± 0.1|1.8 ± 1.2|106|
> |1 |1.1 ± 0.1|1.7 ± 0.6|125|
> |5 (default)|1.2 ± 0.1|1.4 ± 0.6|201|
> |10|1.1 ± 0.2|1.2 ± 0.7|295|
>
> \* Simultaneous: auxiliary and main models updated jointly without inner loops
>
> **Directional KL Components ($\lambda=0.4, K=5$)**
>
> |Flow Loss Direction|KL ($t=0.5$)|KL ($t=1.0$)|
> |-|-|-|
> |Forward only (1→2)|1.1 ± 0.1|1.4 ± 0.6|
> |Reverse only (2→1)|1.2 ± 0.1|1.4 ± 0.7|
> |Bidirectional|1.2 ± 0.1|1.4 ± 0.6|
>
> Without flow loss ($\lambda =0$), the model fails to generalise to unobserved regions, with KL $\simeq$ 23 at $t=1.0$. Adding flow loss dramatically improves performance; with $\lambda = 0.2\text{-}0.8$, KL stays around 1.2 across time steps.
>
> Applying flow loss in only one direction already reduces KL divergence. However, since forward KL encourages broad coverage of the distribution, and reverse KL focuses on matching high-density regions, we decided to use both equally to balance these effects with symmetry.
>
> A similar trend was observed on MoCap setup 2 and we will include these ablations in the revision.
> ## Main paper’s content organisation (TqQA, yfA9)
> We will move key architectural details (NSF design, training algorithm) to the main text for clarity.
>
> ---
> # Specific questions from Reviewer aZhw
> Thank you for your valuable feedback. We address your specific concerns below.
> ## Motivation for Bidirectional KL Divergence in Flow Loss
> We appreciate this opportunity to clarify our design choice.
>
> In principle, a flow consistency loss should make sure the agreement between the one-step transitions $p_\theta\left(x_{t_k} \mid x_{t_i}\right)$ and the two-step composition $\int p_\theta\left(x_{t_k} \mid x_{t_j}\right) p_\theta\left(x_{t_j} \mid x_{t_i}\right) d x_{t_j}$ by minimising a statistical divergence/distance between the two. However, the two-step composition involves integration over $x_{t_j}$, which is not tractable. This makes the choice of statistical divergence tricky and motivates our development of the flow loss.
>
> - Why KL divergence: Among many divergence choices, KL divergence is appealing, because by introducing a variational auxiliary distribution $q_\phi\left(x_{t_j} \mid x_{t_i}, x_{t_k}\right)$, we can minimise the upper-bounds of both the forward and reverse KL in a tractable way, and this is the core idea of our flow loss. These bounds only require sampling from $q_\phi$ and evaluating log-densities from the one-step flow, making training stable and low-cost. Other options like Wasserstein distance are computationally expensive, Stein discrepancies require (intractable) score functions and careful tuning, and adversarial methods often lead to unstable training.
> - Why bi-directional KL: Although our ablation study shows that using forward- or reverse-only KL can already improve performance, we argue that combining both directions with equal weight is a more principled choice. The forward KL encourages the one-step transition to cover the support of the two-step composition, and vice versa. Importantly, both losses share the same auxiliary distribution $q_\phi\left(x_{t_j} \mid x_{t_i}, x_{t_k}\right)$, so optimising either direction helps refine $q_\phi$, and improvements in $q_\phi$ in turn tighten the bound in both directions. This can stabilise training and encourages $q_\phi$ to approximate the true bridge distribution more efficiently. Moreover, the symmetric formulation avoids introducing directional bias into the consistency regularisation.
> ## Ablation Studies on Regularization Term
> Please see “Hyperparameter Sensitivity and Ablation Studies” in common response.
> ## SDE Matching Comparison (Table 2)
> SDE Matching's code was not publicly available at NeurIPS submission time, and their paper didn’t have sufficient implementation details for reproduction. After submission deadline, SDE Matching paper authors have since released their Stochastic Lorenz attractor implementation. Still SDE Matching’s code for other experiments is not available.
>
> We have now conducted the comparison on Stochastic Lorenz attractor. Overall our method achieves better KL divergence with more than 2 orders of magnitude fewer FLOPs.
> |**Method**|**$t=0.25$**||**$t=0.5$**||**$t=0.75$**||**$t=1.0$**||
> |-|-|-|-|-|-|-|-|-|
> ||KL|kFLOPs|KL|kFLOPs|KL|kFLOPs|KL|kFLOPs|
> |**SDE Matching**|||||||||
> |$\Delta t=0.0001$|4.3 ± 0.7|184338|5.3 ± 1.0|368677|3.4 ± 0.8|553015|3.8 ± 1.0|737354|
> |$\Delta t=0.01$|6.3 ± 0.4|1880|11.7 ± 0.5|3760|7.9 ± 0.3|5640|6.0 ± 0.3|7520|
> |**NSF (ours)**|||||||||
> |$t_\mathrm{max}=1.0$|0.8 ± 0.7|53|1.3 ± 0.1|53|0.6 ± 0.3|53|0.2 ± 0.6|53|
> |$t_\mathrm{max}=0.5$|2.4 ± 1.9|53|1.3 ± 0.1|53|1.0 ± 0.4|105|1.7 ± 1.1|105|
> |$t_\mathrm{max}=0.25$|1.2 ± 0.7|53|1.2 ± 0.1|105|0.8 ± 0.4|156|1.3 ± 0.8|208|
>
> We also tried to extend their Stochastic Lorenz implementation to Motion Capture, but it did not reproduce their reported scores. We have contacted the authors, they confirmed that an official implementation for Motion Capture experiment will be released soon, but not before the NeurIPS author feedback period ends. Hopefully we can include comprehensive SDE Matching comparisons for all tasks in our camera-ready version. If their Motion Capture implementation is still unavailable by camera-ready deadline, we will provide results using our best-effort reproduction with appropriate caveats about potential implementation differences.
>
> ---
> ## References
> For 1-50, please refer to the main text.
>
> [51] Zhai et al., "Normalizing Flows are Capable Generative Models", ICML 2025.
>
> ------
> Thank you again, and we hope this addresses your concerns and would be happy to provide any additional clarifications.

---

> > ### Comment · Reviewer_aZhw · 2025-08-04
> >
> > I thank the authors for their detailed response, which effectively addressed my main concerns. In particular, the new experimental results clearly demonstrate the importance of the proposed auxiliary loss (kinda in any form). Additionally, the authors made a commendable effort to provide a fair comparison with SDE matching and I will hence increase my score.

---

> > > ### Author Response · Authors · 2025-08-05
> > >
> > > Thank you for your follow-up and for increasing your score.
> > >
> > > In the camera-ready version, we will (i) expand the explanation and intuition behind the proposed flow-consistency loss (including why bi-directional KL is practical here), (ii) include the ablation studies (no-regularisation, forward-only, reverse-only, and $\lambda$ sweeps) in the appendix with training-time overheads, and (iii) include the SDE Matching comparisons we described.
> > >
> > > In addition, we will incorporate all clarifications, analyses, and ablation results provided in our rebuttal into the camera-ready version.
> > >
> > > We appreciate your constructive suggestions, which materially improve our paper.

---

### Official Review · Reviewer_YJDb · 2025-07-02

**Clarity:** 3
**Significance:** 3
**Originality:** 3
**Rating:** 5
**Confidence:** 2

**Summary:**

The paper proposes neural stochastic flows, which learn transition laws via conditional normalizing flows. The normalizing flows are constructed in such a way that the identity and Markov properties are respected and trained with an additional regularization loss for the flow property/Chapman Kolmogorov equation. The model is tested on three experiments, improving over baselines in terms of speed and performance.

**Questions:**

- The way $x_{t_i}$ is sampled in the expectations in eq. (8) and eq. (10) is a bit unclear. Is it just obtained as a random sample from the dataset at time $t_i$?
- l238: "this is due to the inductive bias of SDEs for better generalization" What is meant by that?

**Ethical Concerns:**

["NO or VERY MINOR ethics concerns only"]

**Final Justification:**

The authors have addressed my concerns and questions in the rebuttal. I maintain my positive rating.

**Limitations:**

yes

**Paper Formatting Concerns:**

no formatting concerns

**Quality:**

3

**Strengths And Weaknesses:**

Strengths:
- The paper is well written and the method seems mathematically sound
- Neural stochastic flows show strong empirical performance while being much faster than non solver-free methods.

Weaknesses:
- A lot of the desired properties of the flows are incentivized through additional loss terms leading to multiple hyperparameters (K, $\lambda$, $\lambda_{\text{1to2}}$, $\lambda_{\text{2to1}}$, $\beta$, $\beta_{\text{skip}}$). It is not clear how to best select all these parameters making extensive hyperparameter tuning necessary. How sensitive is the method to the choice of hyperparameters in practice?
- The dependence on affine coupling layers makes the method not very scalable. Is there a way to lift this dependence?

Minor:
- l26: before consistency models, general distillation approaches for diffusion models also tried to achieve this using the probability flow ODE to define the mapping [1]
- l130: "regularisaion" -> regularisation

I am not familiar enough with the recent related work to properly assess the novelty.

[1] Progressive Distillation for Fast Sampling of Diffusion Models https://openreview.net/pdf?id=TIdIXIpzhoI

---

> ### Author Rebuttal · Authors · 2025-07-30
>
> We thank all reviewers for constructive feedback for which we will include in revision. We are encouraged that reviewers recognise our work as addressing an important problem with a novel approach. Below we will first state our core contributions and address common concerns, and then answer reviewer specific questions.
> # Summary of contributions
> We propose Neural Stochastic Flows (NSF) as a solver-free method to directly learns the weak solution of an SDE from data. Key components include
>
> - Introduction of stochastic flows to the field of sequential deep generative models, enabling one-step computation of conditional distributions of an SDE at arbitrary time points,
> - A conditional normalisation flow that preserves important properties of stochastic flows,
> - The “flow loss” as a regulariser to enforce flow consistency.
>
> Reviewers TqQA and YJDb endorsed the novelty of our contributions on one-step SDE conditional distribution evaluations, noting NSF’s mathematical soundness and substantial speedups. When compared with baselines, Reviewer aZhw commended on NSF being significantly faster while achieving comparable performance regarding accuracies.
> # Common questions
> ## Scalability (YJDb, yfA9)
> Regarding computation, our architecture is based on affine coupling-based flows, which is scalable, e.g., our stochastic moving MNIST experiment is conducted on videos with frame size 64 x 64, and RealNVP [12] and Glow [27] (based on affine coupling flows) has been applied to images. Future work will incorporate our proposed constraints into more expressive normalising flow architectures, e.g., TarFlow [51].
> ## Clarifying inductive bias of SDEs in MoCap dataset (YJDb, yfA9)
> Nonlinear SDEs naturally capture real-world stochastic dynamics in human motion, where random perturbations interact nonlinearly with the system state. This generalises better than ODEs (deterministic) and linear SDEs (state‑independent noise).
>
> Among nonlinear SDE approaches, Latent SDE models rely on controlled SDEs to represent the posterior, which involves learning complex drift function, that is difficult to optimise. SDE Matching addresses challenges by learning posteriors conditioned on the observations, $p(x_t \mid o_{t_1}, o_{t_2}, \ldots)$, but must learn complex, time-varying distributions. In contrast, our Latent NSF takes a simpler yet effective way: directly modeling transition laws $p(x_{t_j}\mid x_{t_i})$, using normalising flows. This avoids needing both control-based SDEs and complex time-dependent posteriors.
> ## Hyperparameter Sensitivity and Ablation Studies (YJDb, aZhw)
> We conducted ablation studies on challenging partial observation tasks. We trained on Stochastic Lorenz Attractor data where only a random continuous segment of length 0.5 from the full trajectory $[0,1]$ was observed.
>
> **Flow Loss Ablation on Missing Stochastic Lorenz**
> |Flow Loss Weight ($\lambda$)|KL ($t=0.5$)|KL ($t=1.0$)|
> |-|-|-|
> |0.0 (no flow loss)|7.9 ± 0.2|22.9 ± 1.8|
> |0.001|3.0 ± 0.2|4.5 ± 0.9|
> |0.01|1.7 ± 0.1|2.1 ± 0.6|
> |0.4 (default)|1.2 ± 0.1|1.2 ± 1.0|
> |0.8|1.3 ± 0.2|1.3 ± 0.6|
>
> **Effect of Auxiliary Optimisation Steps ($\lambda=0.4$)**
> |K (auxiliary steps)|KL ($t=0.5$)|KL ($t=1.0$)|Epoch Time (s)|
> |-|-|-|-|
> |0 (no auxiliary)|7.9 ± 0.2|22.9 ± 1.8|66|
> |0 (simultaneous)*|1.1 ± 0.1|1.8 ± 1.2|106|
> |1 |1.1 ± 0.1|1.7 ± 0.6|125|
> |5 (default)|1.2 ± 0.1|1.4 ± 0.6|201|
> |10|1.1 ± 0.2|1.2 ± 0.7|295|
>
> \* Simultaneous: auxiliary and main models updated jointly without inner loops
>
> **Directional KL Components ($\lambda=0.4, K=5$)**
>
> |Flow Loss Direction|KL ($t=0.5$)|KL ($t=1.0$)|
> |-|-|-|
> |Forward only (1→2)|1.1 ± 0.1|1.4 ± 0.6|
> |Reverse only (2→1)|1.2 ± 0.1|1.4 ± 0.7|
> |Bidirectional|1.2 ± 0.1|1.4 ± 0.6|
>
> Without flow loss ($\lambda =0$), the model fails to generalise to unobserved regions, with KL $\simeq$ 23 at $t=1.0$. Adding flow loss dramatically improves performance; with $\lambda = 0.2\text{-}0.8$, KL stays around 1.2 across time steps.
>
> Applying flow loss in only one direction already reduces KL divergence. However, since forward KL encourages broad coverage of the distribution, and reverse KL focuses on matching high-density regions, we decided to use both equally to balance these effects with symmetry.
>
> A similar trend was observed on MoCap setup 2 and we will include these ablations in the revision.
> ## Main paper’s content organisation (TqQA, yfA9)
> We will move key architectural details (NSF design, training algorithm) to the main text for clarity.
>
> ---
> # Specific questions from Reviewer YJDb
>
> Thank you for your thoughtful review and constructive feedback. We appreciate your recognition of our work's mathematical soundness and strong empirical performance. Below we address your concerns.
>
> ## Hyperparameter sensitivity
>
> We understand your concern about multiple hyperparameters. Please see "Hyperparameter Sensitivity and Ablation Studies" above, where we present comprehensive ablation results demonstrating the method's robustness.
>
> ## Scalability and Architecture
>
> Please see “Scalability” above in common response, where we discuss the limitation on the scalability and architecture.
>
> ## Sampling procedure for expectation
>
> Thank you for your careful reading and specific questions.
> The sampling procedure in Eq. 8 corresponds to the nested expectation $\mathbb{E}\_{ p\_\theta(x_{t_k} | x_{t_i}) } \left[ \mathbb{E}\_{q_\phi (x_{t_j} |x_{t_i}, x_{t_k}) }[ \cdot ]\right]$.
> We first sample the initial state $x_{t_i}$ from the dataset. Given a time triplet $(t_i, t_j, t_k)$ sampled according to the strategy described in Appendix l.868, we sample $x_{t_k} \sim p_\theta\left(x_{t_k} \mid x_{t_i}\right)$ using the neural stochastic flow. Then, conditioned on $x_{t_i}$ and $x_{t_k}$, we sample $x_{t_j} \sim q_\phi\left(x_{t_j} \mid x_{t_i}, x_{t_k}\right)$ from the auxiliary bridge distribution.
>
> Similarly, this expectation in Eq. 10 corresponds to the nested expectation $\mathbb{E}\_{p\_\theta(x_{t_j} \mid x_{t_i})}\left[\mathbb{E}\_{p\_\theta(x_{t_k} \mid x_{t_j})}[\cdot]\right]$.
> We first sample the initial state $x_{t_i}$ from the dataset. Then, using our neural stochastic flow (NSF) model, we sample $x_{t_j} \sim p_\theta\left(x_{t_j} \mid x_{t_i}\right)$, and subsequently sample $x_{t_k} \sim p_\theta\left(x_{t_k} \mid x_{t_j}\right)$ using the same model.
>
> We will clarify these expectation in camera-ready version.
>
> ## Inductive bias of SDEs
>
> Please see “Clarifying inductive bias of SDEs in MoCap dataset” above in common response.
>
> ## Minor Points
>
> Thank you for the reference to Progressive Distillation. We'll add this important connection and will also fix the typo on l130.
>
> ---
> ## References
> For 1-50, please refer to the main text.
>
> [51] Zhai et al., "Normalizing Flows are Capable Generative Models", ICML 2025.
>
> ---
>
> We hope this addresses your concerns. We're committed to making these clarifications prominent in the final version to help future practitioners effectively use our method.

---

> > ### Comment · Area_Chair_8vQr · 2025-08-05
> >
> > Dear reviewer YJDb,
> >
> > After considering the rebuttal to your review and the other reviews/rebuttals, how and why has this affected your position on this submission? Please reply with an official comment (not just the mandatory acknowledgement) reflecting your current view, any follow-up questions/comments etc.
> >
> > Note the Aug 6 AoE deadline, make sure to respond in time for the authors to be able to submit a response if necessary.

---

> > > ### Comment · Reviewer_YJDb · 2025-08-05
> > >
> > > Thank you for answering my questions. I'm still not convinced of the scalability and think that the number of hyperparameters/finetuning is a weakness of the approach, but I don't see them as critical. I will keep my current score (5).

---

> > > > ### Author Response · Authors · 2025-08-07
> > > >
> > > > Thank you for carefully considering our rebuttal and for maintaining your positive assessment. We appreciate your constructive feedback throughout the process.
> > > >
> > > > We acknowledge the concerns about scalability and hyperparameter complexity. In the camera-ready version, we will add practical guidance (default ranges and tuning tips) for key hyperparameters in the appendix. Regarding scalability, we will clarify that our coupling-flow backbone inherits scalability from RealNVP/Glow-style architectures, though such affine-coupling designs can show limitations in generative quality for high-resolution data. As a direction for future work, we will also mention the possibility of applying our flow-consistency constraints to more expressive architectures such as TarFlow.
> > > >
> > > > We will incorporate all clarifications, analyses, and ablations from the rebuttal into the camera-ready version. Thank you again for helping improve the paper.

---

### Official Review · Reviewer_TqQA · 2025-07-03

**Clarity:** 4
**Significance:** 4
**Originality:** 4
**Rating:** 5
**Confidence:** 3

**Summary:**

The paper focuses on using neural stochastic flows to model stochastic dynamic data without solving SDEs. To do this, it is proposed to use conditional normalising flows to learn the SDE transition distribution. It also allows efficient one-step sampling to facilitate the computational efficiency. Experiments were conducted with Stochastic Lorenz Attractor, CMU Motion Capture, and Stochastic Moving MNIST.

**Questions:**

Algorithm in Appendix C can be written in the main text, since it helps to better follow the process. Minor: (1) line 130: typo in "regularisation"; (2) Line 170: in the description of the total objective, "the flow loss" might refer to the Eq (25) in the appendix instead of Eq (24) as in the current version.

**Ethical Concerns:**

["NO or VERY MINOR ethics concerns only"]

**Final Justification:**

Dear Area Chairs,

In the rebuttal phase, authors have discussed and adequately addressed specific questions and suggestions in my reviews of the paper. For the final justification, I maintain my overall score of 5: Accept.

Kind regards

**Limitations:**

Yes

**Quality:**

4

**Strengths And Weaknesses:**

* The work focused on important problem. To avoid learning the transition law via simulations with a large amount of steps, the paper follows a solver-free approach. The paper proposed a specific design (architecture) for parameterising the conditional normalising flows, and the loss to train this. To capture the noisy scenarios, the framework is also extended to Latent-NFS.

* The paper is written in self-contained way. Figure 1 can well-illustrate the concept behind the proposed approach (neural stochastic flows-NSF). Section 2 briefly introduce to the SDEs and Stochastic Flows to follow the next sections of the paper.

* The approach (NSF and Latent NSF) has strong motivations and technically sound.
   - To avoid costly computation of characterising strong solutions of SDEs, the approach derives the alternative weak solutions. A parametric model is then proposed to approximate it (Eq (2) and (3), with design in section 3.1). The conditions for the converted weak solutions of SDEs are described provided in section 3.
   - To enforce the flow property (especially the property 2 in Section 3) in NSFs, the approach utilised variational technique to construct upper-bounds on KL divergence for $\mathcal{L}_{\text{flow}}$ beside the normal supervised loss.
   - The connection made for the choice of the structure of Latent NSF is described in lines 157-164 with support in the Figure 2.

* The experiments is showed that the computational cost is significantly reduced, which is an important contribution in term of the efficiency of this approach. Compared to other Latent- and Neural- SDE existing approaches, NSF obtained low KL divergence (high performance) with much lower computational cost (Table 1). Analyses about the trade-off between computation and performance is also conducted in Figure 4 (with CMU Motion Capture dataset) and Table 3 (for Stochastic Moving MNIST).

---

> ### Author Rebuttal · Authors · 2025-07-30
>
> We thank all reviewers for constructive feedback for which we will include in revision. We are encouraged that reviewers recognise our work as addressing an important problem with a novel approach. Below we will first state our core contributions and address common concerns, and then answer reviewer specific questions.
> # Summary of contributions
> We propose Neural Stochastic Flows (NSF) as a solver-free method to directly learns the weak solution of an SDE from data. Key components include
>
> - Introduction of stochastic flows to the field of sequential deep generative models, enabling one-step computation of conditional distributions of an SDE at arbitrary time points,
> - A conditional normalisation flow that preserves important properties of stochastic flows,
> - The “flow loss” as a regulariser to enforce flow consistency.
>
> Reviewers TqQA and YJDb endorsed the novelty of our contributions on one-step SDE conditional distribution evaluations, noting NSF’s mathematical soundness and substantial speedups. When compared with baselines, Reviewer aZhw commended on NSF being significantly faster while achieving comparable performance regarding accuracies.
> # Common questions
> ## Scalability (YJDb, yfA9)
> Regarding computation, our architecture is based on affine coupling-based flows, which is scalable, e.g., our stochastic moving MNIST experiment is conducted on videos with frame size 64 x 64, and RealNVP [12] and Glow [27] (based on affine coupling flows) has been applied to images. Future work will incorporate our proposed constraints into more expressive normalising flow architectures, e.g., TarFlow [51].
> ## Clarifying inductive bias of SDEs in MoCap dataset (YJDb, yfA9)
> Nonlinear SDEs naturally capture real-world stochastic dynamics in human motion, where random perturbations interact nonlinearly with the system state. This generalises better than ODEs (deterministic) and linear SDEs (state‑independent noise).
>
> Among nonlinear SDE approaches, Latent SDE models rely on controlled SDEs to represent the posterior, which involves learning complex drift function, that is difficult to optimise. SDE Matching addresses challenges by learning posteriors conditioned on the observations, $p(x_t \mid o_{t_1}, o_{t_2}, \ldots)$, but must learn complex, time-varying distributions. In contrast, our Latent NSF takes a simpler yet effective way: directly modeling transition laws $p(x_{t_j}\mid x_{t_i})$, using normalising flows. This avoids needing both control-based SDEs and complex time-dependent posteriors.
> ## Hyperparameter Sensitivity and Ablation Studies (YJDb, aZhw)
> We conducted ablation studies on challenging partial observation tasks. We trained on Stochastic Lorenz Attractor data where only a random continuous segment of length 0.5 from the full trajectory $[0,1]$ was observed.
>
> **Flow Loss Ablation on Missing Stochastic Lorenz**
> |Flow Loss Weight ($\lambda$)|KL ($t=0.5$)|KL ($t=1.0$)|
> |-|-|-|
> |0.0 (no flow loss)|7.9 ± 0.2|22.9 ± 1.8|
> |0.001|3.0 ± 0.2|4.5 ± 0.9|
> |0.01|1.7 ± 0.1|2.1 ± 0.6|
> |0.4 (default)|1.2 ± 0.1|1.2 ± 1.0|
> |0.8|1.3 ± 0.2|1.3 ± 0.6|
>
> **Effect of Auxiliary Optimisation Steps ($\lambda=0.4$)**
> |K (auxiliary steps)|KL ($t=0.5$)|KL ($t=1.0$)|Epoch Time (s)|
> |-|-|-|-|
> |0 (no auxiliary)|7.9 ± 0.2|22.9 ± 1.8|66|
> |0 (simultaneous)*|1.1 ± 0.1|1.8 ± 1.2|106|
> |1 |1.1 ± 0.1|1.7 ± 0.6|125|
> |5 (default)|1.2 ± 0.1|1.4 ± 0.6|201|
> |10|1.1 ± 0.2|1.2 ± 0.7|295|
>
> \* Simultaneous: auxiliary and main models updated jointly without inner loops
>
> **Directional KL Components ($\lambda=0.4, K=5$)**
>
> |Flow Loss Direction|KL ($t=0.5$)|KL ($t=1.0$)|
> |-|-|-|
> |Forward only (1→2)|1.1 ± 0.1|1.4 ± 0.6|
> |Reverse only (2→1)|1.2 ± 0.1|1.4 ± 0.7|
> |Bidirectional|1.2 ± 0.1|1.4 ± 0.6|
>
> Without flow loss ($\lambda =0$), the model fails to generalise to unobserved regions, with KL $\simeq$ 23 at $t=1.0$. Adding flow loss dramatically improves performance; with $\lambda = 0.2\text{-}0.8$, KL stays around 1.2 across time steps.
>
> Applying flow loss in only one direction already reduces KL divergence. However, since forward KL encourages broad coverage of the distribution, and reverse KL focuses on matching high-density regions, we decided to use both equally to balance these effects with symmetry.
>
> A similar trend was observed on MoCap setup 2 and we will include these ablations in the revision.
> ## Main paper’s content organisation (TqQA, yfA9)
> We will move key architectural details (NSF design, training algorithm) to the main text for clarity.
>
> ---
> # Specific questions from Reviewer TqQA
>
> We sincerely thank you for your thorough review and positive assessment of our work. We are delighted that you found our approach technically sound and appreciate your recognition of our contributions to computational efficiency in SDE modeling. Below we address your specific questions:
>
> ## Algorithm Placement
>
> We appreciate your suggestion to move the training algorithm from Appendix C to the main text, as we wrote in “Main paper’s content organisation”. We agree this would improve readability and help readers better follow the training process.
>
> ## Minor Corrections
>
> Thank you for catching typo and wrong reference. We will fix them in the camera-ready version.
>
> ---
> ## References
> For 1-50, please refer to the main text.
>
> [51] Zhai et al., "Normalizing Flows are Capable Generative Models", ICML 2025.
>
> ------
> Thank you again, and we hope this addresses your concerns and would be happy to provide any additional clarifications.

---

> > ### Comment · Area_Chair_8vQr · 2025-08-05
> >
> > Dear reviewer TqQA,
> >
> > After considering the rebuttal to your review and the other reviews/rebuttals, how and why has this affected your position on this submission? Please reply with an official comment (not just the mandatory acknowledgement) reflecting your current view, any follow-up questions/comments etc.
> >
> > Note the Aug 6 AoE deadline, make sure to respond in time for the authors to be able to submit a response if necessary.

---

> ### Comment · Reviewer_TqQA · 2025-08-05
> **Response to Authors**
>
> Dear Authors,
>
> Thank you for your great effort to respond and have further detail discussions about the approach. The responses adequately address my questions and comments in the review.
>
> Kind regards

---

> > ### Author Response · Authors · 2025-08-07
> >
> > Thank you for your positive evaluation and constructive feedback throughout the review process. We greatly appreciate your recognition of our work’s contributions and your score of 5.
> >
> > We are pleased that our rebuttal addressed your questions, and we are grateful to improve the paper in the camera-ready version based on your suggestions.
> >
> > We will incorporate all suggested improvements to ensure clarity and reproducibility. Thank you again for helping us strengthen the paper.

---

### Note · Authors · 2025-08-12

We thank the reviewers for their constructive discussion. Reviewers recognised the core novelty: Neural Stochastic Flows (NSF) learn SDE transition laws with normalising flows, enabling one-step sampling at arbitrary time gaps while preserving identity/Markov structure and achieving large speed-ups with comparable accuracy (TqQA, YJDb, aZhw), offering theoretical soundness (YJDb, yfA9).

During rebuttal we provided evidence in respond to reviewers’ concerns. In the ablation study we showed that flow-consistency loss is essential: without it, errors on the missing Lorenz task grow sharply, whereas a modest weight keeps KL being around 1.2 across horizons; forward-only and reverse-only variants also help; we showed the effect on run-time and accuracy w.r.t. the number of steps for auxiliary distribution training. We have result with the similar trend on MoCap, which will be on the camera-ready.

- These study addressed requests for empirical justification and were acknowledged in the follow-ups (TqQA, aZhw, YJDb).

We also showed a comparison to SDE Matching using newly released code: on stochastic Lorenz, NSF attains better KL with $10^2$–$10^3$ times fewer FLOPs; we will include full protocols (aZhw). We clarified the sampling in Eqs. (8,10), the meaning of t, configuration for MoCap, how Fig. 3 was generated, and known reproducibility issues for Latent SDE.

- Reviewers agreed these points were resolved (TqQA, yfA9). Concerns about scalability and hyperparameters were noted but not considered critical (YJDb), and further guidance on hyperpatameters was requested (yfA9); we will include this guidance in the camera-ready.

If the paper is accepted, then for the camera-ready, we will reflect all the points we have discussed. For example, we will move the training algorithm and architecture to the main text; expand the rationale for the consistency objective (why KL, why bi-directional); add practical defaults and sensitivity guidance for hyperpatameters; report parameter counts and training-time overheads; include SDE Matching; and discuss scalability and applicability to more expressive flows, such as TarFlow.

In summary, NSF provides accurate one-step conditional sampling across arbitrary time gaps with dramatic computational savings, and the consistency objective materially improves generalisation under various settings. The rebuttal clarifications and the above commitments address the concerns and further strengthen the paper (TqQA, YJDb, aZhw, yfA9).

---

### Decision · Program_Chairs · 2025-09-17

**Decision:**

Accept (poster)

**Comment:**

This submission introduces a new view of continuous-time dynamics modeling based on approximating a stochastic flow, the transition probabilities of a stochastic process. The approach relies on coupling-style normalizing flows that by definition satisfies 3 of the 4 standard properties of a stochastic flow. The flow property is satisfied approximately by combining a cross-entropy/likelihood objective with a regularisation loss. Neural tochasic flow can be thought of as a stochastic analogue to the deterministic distillation-like techniques such as consistency trajectory models, flow map matching, etc.

The reviewers reached a consensus that this is an interesting approach for time-series modeling that is theoretically sound and potentially practically useful and thus I recommend acceptance. The main concerns were regarding experiments and significant amount of hyperparameters which the authors mostly resolved during the rebuttal phase. When revising the paper, please pay attention to the promised improvements and extra results. I also strongly urge the authors to include both the original result of the Li et al. (2020) paper as well as the failed replication.